# Learning Large-Scale MTP$_2$ Gaussian Graphical Models via Bridge-Block Decomposition

**Xiwen Wang**[1], **Jiaxi Ying**[1,2]*, **Daniel P. Palomar**[1]

The Hong Kong University of Science and Technology[1]

HKUST Shenzhen-Hong Kong Collaborative Innovation Research Institute[2]

{xwangew, jx.ying}@connect.ust.hk, palomar@ust.hk

## Abstract

This paper studies the problem of learning the large-scale Gaussian graphical models that are multivariate totally positive of order two (MTP$_2$). By introducing the concept of bridge, which commonly exists in large-scale sparse graphs, we show that the entire problem can be equivalently optimized through (1) several smaller-scaled sub-problems induced by a *bridge-block decomposition* on the thresholded sample covariance graph and (2) a set of explicit solutions on entries corresponding to *bridges*. From practical aspect, this simple and provable discipline can be applied to break down a large problem into small tractable ones, leading to enormous reduction on the computational complexity and substantial improvements for all existing algorithms. The synthetic and real-world experiments demonstrate that our proposed method presents a significant speed-up compared to the state-of-the-art benchmarks.

## 1 Introduction

In recent decades, the surge in data availability has posed challenges for analyzing and understanding large-scale datasets. A key challenge is examining pairwise relationships among numerous variables, which can be represented using Gaussian graphical models (GGMs) [1] that depict variable connections as graphs. The precision matrix, which is the inverse of the covariance matrix, helps determine the non-zero patterns of GGMs [2, 3]. A traditional approach to estimate the precision matrix is the graphical lasso [4, 5], which is formulated as a regularized log-determinant program. The solutions of graphical lasso possess an appealing property known as sparsity, which severs as a common assumption particularly in large graphical models and has been shown to offer numerous benefits by a significant amount of research. For example, the sparsity can reduce the model size and improve the interpretability of the model by highlighting the most important variables and their relationships, allowing better understanding the underlying causal structure [6–8].

In this paper, we solve the large-scale sparse precision matrix estimation problem for GGMs that follow a multivariate totally positive of order two (MTP$_2$) Gaussian distribution [9, 10], or equivalently, possess a precision matrix that is a symmetric $M$-matrix [11], where all off-diagonal elements are non-positive. The so-called MTP$_2$ property is a special form of positive dependence and plays an essential role in applications where all interaction potentials are considered non-negative or the focus is on emphasizing positive associations between variables [12–15]. The MTP$_2$ property has been applied in various fields. Here are some examples. In finance, MTP$_2$ structures are often exploited as the asset returns are often considered positively correlated [16–18]; In machine learning, MTP$_2$ GGMs are recognized as attractive Markov random field and have been used in applications such as taxonomic reasoning [19] and psychometrics [13]. The MTP$_2$ precision matrix, also referred to

---

*Corresponding author.

37th Conference on Neural Information Processing Systems (NeurIPS 2023).

as the generalized graph Laplacian [20–22], along with the combinatorial graph Laplacian [23–27], have found broad applications across a variety of fields due to their unique characteristics. These applications include, but are not limited to, graph Fourier transform [28], electrical circuits analysis [29], image and video coding [30], financial time-series analysis [31, 32], and structured graph learning [33, 34].

The objective of this paper is to build a theoretical foundation and devise effective approaches for learning large-scale, sparse $MTP_2$ GGMs. The contributions of this paper are threefold.

- This is the first work in the literature that introduces the notion of bridges in the context of learning $MTP_2$ GGMs to the best of our knowledge. Building upon this notion, we prove that the presence of explicit solutions for some entries depends on whether an edge functions as a bridge. Meanwhile, we demonstrate that the optimal solution exhibits a decomposed structure through a vertex-partition known as bridge-block decomposition.

- Based on theoretical findings, we propose an efficient framework that decomposes a large problem into several small tractable ones, each of which can be solved using any existing algorithm. With some negligible extra cost for bridge-block decomposition, the proposed method results in a dimension reduction that significantly cuts down the computational complexity.

- The synthetic and real-world experiments demonstrate that our proposed methods prompt a considerable speed-up for all existing methods and enable the solving of large-scale $MTP_2$ GGMs that were previously considered infeasible.

### 1.1   Notation and Organization

Vectors and matrices are written as lower and upper case bold letters, respectively. An undirected graph is denoted as $G = (\mathcal{V}, \mathcal{E})$, where $\mathcal{V}$ is the set of nodes with size $|\mathcal{V}| = p$ and $\mathcal{E} \subset \mathcal{V} \times \mathcal{V}$ is the set of edges. Note that we shall not distinguish between $(i, j) \in G$ and $(i, j) \in \mathcal{E}$. Some graph terminology frequently used throughout the paper are introduced as follows.

- **Support graph** Given a symmetric matrix $\boldsymbol{A} \in \mathbb{S}^p$, the support graph of $\boldsymbol{A}$, denoted as supp $(\boldsymbol{A})$, is defined as an undirected graph with the vertex set $\mathcal{V} = \{1, \dots, p\}$ and the edge set $\mathcal{E} \in \mathcal{V} \times \mathcal{V}$ such that $(i, j) \in \mathcal{E}$ if and only if $A_{ij} \neq 0$ for every two different vertices $i, j \in \mathcal{V}$.

- **Neighbors** The neighbors of a node $i$ refer to the subset of vertices that are connected to this node, i.e., $\mathcal{N}(i) = \{j \mid (i, j) \in \mathcal{E}\}$.

- **Path and Cycle** A path from node $i_1 \in \mathcal{V}$ to node $i_T \in \mathcal{V}$ is a sequence (i.e., an ordered set) of edges denoted as $d_{i_1, i_T}$, each one incident to the next, i.e., $\{(i_t, i_{t+1})\}_{t=1}^{T-1} \subseteq \mathcal{E}$. A cycle is a path from a node to itself when it does not duplicately include the same edge.

- **Partition** A partition $\mathcal{P} = \{\mathcal{V}_1, \dots, \mathcal{V}_K\}$ is a collection of non-empty, disjoint vertex sets $\mathcal{V}_1, \dots, \mathcal{V}_K$ called cluster such that $\bigcup_{i=1}^K \mathcal{V}_k = \mathcal{V}$.

- **Component** A component of $G$ refers to a subset of vertices that are connected to each other by edges, but are not connected to any other vertices in the graph.

The paper is organized as follows. In Section 2, we introduce the problem formulation and related works. In Section 3, we present our main result along with its potential impact. Section 4 presents the numerical experiments to show that the proposed methods can facilitate the performance of existing algorithms and solve large-scale $MTP_2$ GGMs problems [2]. In Section 5 we draw the conclusions.

## 2   Problem Formulation and Related Works

### 2.1   Problem Formulation

Let $\boldsymbol{y}$ be a random vector that follows a Gaussian distribution with zero mean and a precision matrix $\boldsymbol{\Theta}$, a.k.a. the inverse of covariance matrix $\boldsymbol{\Sigma}$. A Gaussian graphical model represents the conditional dependency between random variables via a graph $G$, in which nodes correspond to $\boldsymbol{y}$ and edges between these variables represent conditional dependencies, which can be equivalently characterized by the non-zero patterns of the precision matrix $\boldsymbol{\Theta}$ [2].

---

[2]Codes are available in `https://github.com/Xiwen1997/mtp2-bbd`.

This paper considers estimating the precision matrix $\boldsymbol{\Theta}$ given $n$ independent and identically distributed observations $\{\boldsymbol{y}_1, \ldots, \boldsymbol{y}_n\}$ that follow an MTP$_2$ Gaussian distribution. Equivalently, $\boldsymbol{\Theta}$ is assumed to be a symmetric $M$-matrix, i.e., $\Theta_{ij} \leq 0$ for any $i \neq j$. The problem is formulated as

$$\underset{\boldsymbol{\Theta} \in \mathcal{M}^p}{\text{minimize}} \quad -\log \det(\boldsymbol{\Theta}) + \langle \boldsymbol{\Theta}, \boldsymbol{S} \rangle + \sum_{i \neq j} \Lambda_{ij} |\Theta_{ij}|, \tag{1}$$

where $\boldsymbol{S}$ is the sample covariance matrix, $\Lambda_{ij} \geq 0$ are the regularization coefficients, the objective is to minimize the negative log-likelihood of the data subject to a weighted $\ell_1$-norm penalty on the precision matrix, and $\mathcal{M}^p$ refers to a set of $M$-matrices with dimension $p$, i.e.,

$$\mathcal{M}^p = \left\{ \boldsymbol{\Theta} \in \mathbb{S}^p \,\middle|\, \boldsymbol{\Theta} \succ \boldsymbol{0} \text{ and } \Theta_{ij} \leq 0, \forall i \neq j \right\}. \tag{2}$$

We refer to $\boldsymbol{\Theta}^\star$ as the optimal solution of (1) and to the support graph of $\boldsymbol{\Theta}^\star$, i.e., $\text{supp}(\boldsymbol{\Theta}^\star)$, as the *optimal graph*. Particularly, we define the *thresholded sample covariance matrix* $\boldsymbol{T}$ as

$$T_{ij} = \begin{cases} S_{ij} - \Lambda_{ij} & \text{if } i \neq j \text{ and } S_{ij} > \Lambda_{ij}, \\ 0 & \text{otherwise,} \end{cases} \tag{3}$$

and the support graph of $\boldsymbol{T}$ as the *thresholded sample covariance graph* (short for *thresholded graph*). Clearly, entries corresponding to $S_{ij} \leq \lambda_{ij}$ will be thresholded to zero. Since the thresholded graph holds immense importance throughout the paper, unless explicitly mentioned otherwise, we denote $G = \text{supp}(\boldsymbol{T})$.

The involvement of MTP$_2$ constraints, which require that $\Theta_{ij} \leq 0$ for all $i \neq j$, confers several advantages [11]. For example, all $M$-matrices are inverse-positive, i.e., $\boldsymbol{\Theta}^{-1} \geq \boldsymbol{0}$, and the non-smoothness from $\ell_1$ norms can be eliminated via $\sum_{i \neq j} \Lambda_{ij} |\Theta_{ij}| = -\langle \boldsymbol{\Lambda}, \boldsymbol{\Theta} \rangle$, where $\boldsymbol{\Lambda} = (\Lambda_{ij})$ and $\text{diag}(\boldsymbol{\Lambda}) = \boldsymbol{0}$. Meanwhile, the MTP$_2$ constraints maintain the sparsity in the estimated precision matrix as an implicit regularizer [19]. Throughout the paper, we require the following assumption.

**Assumption 2.1.** For any $i \neq j$, we have $S_{ij} < \sqrt{S_{ii}S_{jj}}$.

Assumption 2.1 holds with probability 1 if the number of observations $n \geq 2$ [13, 19]. Under that assumption, the minimizer of Problem (1) exists and is unique according to [19, Theorem 1].

## 2.2 Related Works

In literature, devising algorithms for learning MTP$_2$ GGMs has garnered considerable attentions. For instance, block coordinate descent (BCD) methods [19, 26, 35] update a single row/column of the precision matrix in a cyclic manner by solving non-negative least squares problems. Projection-based methods, such as projected gradient descent [36] and projected quasi-Newton algorithms [37], iteratively take steps along the descent direction and then project the solutions back to the feasible region. Despite these efforts, existing research has not fully explored the potential of exploiting MTP$_2$ properties to reduce computational complexity. With complexities of $\mathcal{O}(p^4)$ for BCD methods and $\mathcal{O}(p^3)$ for projection-based techniques, addressing large-scale problems continues to be challenging, particularly when manipulating full-dimensional matrices without problem reduction.

Instead of devising efficient algorithms, recently, leveraging the properties of the thresholded sample covariance graph has emerged as a popular approach for learning GGMs. Specifically, the existence of closed-form solution has been established for graphical lasso when the thresholded sample covariance graph is acyclic (i.e., contains no cycles) [38, 39]. However, the applicability of their main results is limited by certain conditions that are difficult to verify. Interestingly, those conditions are unnecessary for the existence of closed-from solutions in MTP$_2$ GGMs [20]. Despite the theoretical establishment on closed-form solutions, an exact acyclic structure is rarely observed in practice. Therefore, more research dives into the relationship between $\text{supp}(\boldsymbol{T})$ and $\text{supp}(\boldsymbol{\Theta}^\star)$. Research found that $\text{supp}(\boldsymbol{\Theta}^\star)$ is a subset of $\text{supp}(\boldsymbol{T})$ [13], the number of connected components in $\text{supp}(\boldsymbol{T})$ is identical to that in $\text{supp}(\boldsymbol{\Theta}^\star)$ [20], and there exist necessary conditions for the presence of edges in $\text{supp}(\boldsymbol{\Theta}^\star)$ via analyzing the path products on thresholded matrix [13].

This paper advances prior research in two ways. Firstly, unlike previous studies that only provided an explicit solution for $\Theta_{ij}$ in the case of acyclic thresholded graphs, we reveal that this explicit solution for $\Theta_{ij}$ consistently applies to all $(i, j)$ pairs acting as bridges, regardless of whether the thresholded graph is acyclic or non-acyclic. Secondly, we highlight that the optimal graph can be represented in an equivalent decomposed form through a vertex partition, termed as the bridge-block decomposition of the thresholded graph by leveraging MTP$_2$ properties.

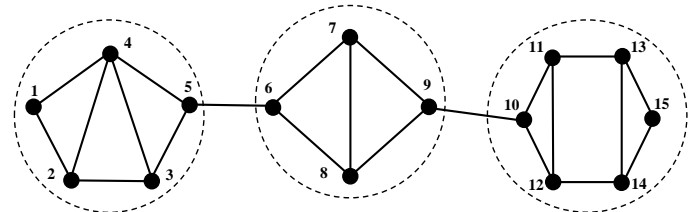

Figure 1: An illustration of the bridge-block decomposition. Edges $(5,6)$ and $(9,10)$ are identified as bridges since removing either of them increases the number of connected components. After removing the bridge edges, the resulting decomposition $\mathcal{P}^{\text{bbd}} = \{\{1,2,3,4,5\}, \{7,8,9,10\}, \{11,12,13,14,15,16\}\}$.

## 3 Proposed Methods

The goal of this paper is not about developing an algorithm but to shed light on the remarkable properties concealed in the thresholded graph. This section is organized as follows. In Section 3.1, we introduce bridge and bridge-block decomposition. Section 3.2 presents our main result. Then in Section 3.3, we elaborate on the contributions and connections to existing research.

### 3.1 Bridge and Bridge-Block Decomposition

Bridge is one of the important concepts in graph theory. Technically, an edge is called a *bridge* if and only if its deletion increases the number of graph components. Therefore, an edge is a bridge only when it is not contained in any cycles [40]. Notably, when a graph consists solely of bridges, it is referred to as *acyclic*. In this paper, we denote $\mathcal{B}$ as the set of all bridges, i.e,

$$\mathcal{B} := \{(i,j) \mid (i,j) \text{ is a bridge in } G\}. \tag{4}$$

*Remark* 3.1. Bridges are frequently observed, particularly in large-scale sparse graphs [41–43]. This is attributed to the fact that in sparse graphs, only the most significant relationships between variables are retained, and the removal of edges can create additional connected components, giving rise to the presence of bridges.

*Remark* 3.2. In practice, the set $\mathcal{B}$ can be efficiently obtained via various bridge-finding algorithms [40, 44]. These algorithms employ a depth-first search approach, resulting in a computational complexity of $\mathcal{O}(|\mathcal{V}| + |\mathcal{E}|)$. In the sparse graphs that are of interest to us, the number of edges $|\mathcal{E}|$ typically scales similarly to the number of nodes $|\mathcal{V}|$. As a result, the computational cost of identifying bridges in large-scale sparse graphs remains low.

Using Figure 1 as an illustration, we perform the *bridge-block decomposition* [45] as follows. By removing all the bridges, we compute the clusters $\mathcal{V}_k$ corresponding to the components of $G = (\mathcal{V}, \mathcal{E} \setminus \mathcal{B})$. This process results in a vertex-partition, known as the bridge-block decomposition:

$$\mathcal{P}^{\text{bbd}} = \{\mathcal{V}_1, \mathcal{V}_2, \ldots, \mathcal{V}_K\}, \tag{5}$$

where $K$ refers to the number of clusters, also the number of components in the graph $(\mathcal{V}, \mathcal{E} \setminus \mathcal{B})$.

Without loss of generality, we define the operator $\psi$ as $\psi(i) = k$ if node $i$ belongs to the $k$-th cluster. In the literature, the bridge-block decomposition is also known as the *2-edge-connected decomposition* [46, 47], a fundamental concept in graph theory with numerous applications such as community search [48], social network mining [49], and transmission networks [50].

In practice, the cost of obtaining the bridge-block decomposition $\mathcal{P}^{\text{bbd}}$, which involves calculating the thresholded graph, bridges, and clusters, is negligible. With the aforementioned preliminary knowledge, we formally introduce our main result as follows.

### 3.2 Main Results

Considering a square matrix $\boldsymbol{A} \in \mathbb{S}^p$, we define $\boldsymbol{A}_{\mathcal{V}_k} \in \mathbb{S}^{p_k}$ as the principal sub-matrix of $\boldsymbol{A}$ keeping the rows and columns indexed by $\mathcal{V}_k$, in which $p_k = |\mathcal{V}_k|$ is the number of nodes in $k$-th cluster and

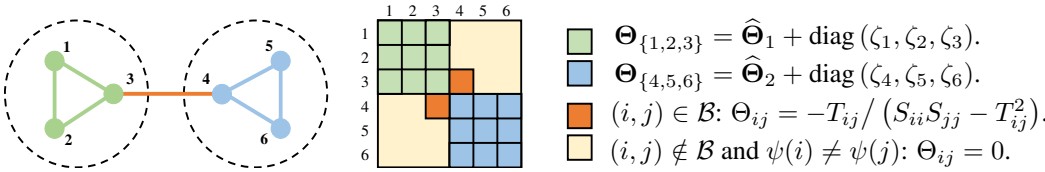

Figure 2: An example of obtaining optimal solution via Theorem 3.3. The thresholded graph of 6 nodes can be partitioned into 2 clusters $\mathcal{V}_1 = \{1, 2, 3\}$ and $\mathcal{V}_2 = \{4, 5, 6\}$. The optimal solution $\Theta \in \mathbb{S}^6$ can be exactly computed via ■ a set of explicit solution $\{\Theta_{3,4}, \Theta_{4,3}\}$, ■■ solutions of smaller-sized sub-problems, i.e., $\widehat{\Theta}_1 \in \mathbb{S}^3$ and $\widehat{\Theta}_2 \in \mathbb{S}^3$, and ▢ zeros in the rest positions.

we have $p = \sum_k p_k$. For each $i \in \mathcal{V}_k$, we mark $\pi(i)$ as its corresponding index in $\mathcal{V}_k$. Hence, we define $\widehat{\Theta}_k$ as the optimal solution of $k$-th sub-problem, i.e.,

$$\widehat{\Theta}_k = \arg \min_{\Theta_k \in \mathcal{M}^{p_k}} - \log \det(\Theta_k) + \langle \Theta_k, S_{\mathcal{V}_k} - \Lambda_{\mathcal{V}_k} \rangle. \tag{6}$$

The main result is then given as follows.

**Theorem 3.3.** *Under Assumption 2.1, given the bridge-block decomposition of the thresholded graph* $\mathrm{supp}(T)$ *as* $\mathcal{P}^{bbd}$, *and the optimal solution of each sub-problem* (46) *as* $\widehat{\Theta}_k$, *the optimal solution of Problem* (1), *i.e.,* $\Theta^\star$, *can be obtained as*

$$\Theta_{i,j}^\star = \begin{cases} [\widehat{\Theta}_k]_{\pi(i),\pi(i)} + \zeta_i & \text{if } i = j \in \mathcal{V}_k, \\ [\widehat{\Theta}_k]_{\pi(i),\pi(j)} & \text{if } i \neq j \text{ and } i, j \in \mathcal{V}_k, \\ -T_{ij}/(S_{ii}S_{jj} - T_{ij}^2) & \text{if } (i,j) \in \mathcal{B}, \\ 0 & \text{otherwise}. \end{cases} \tag{7}$$

*in which* $\zeta_i = \frac{1}{S_{ii}} \sum_{(i,m) \in \mathcal{B}} \frac{T_{im}^2}{S_{ii}S_{mm} - T_{im}^2}$ *and* $\zeta_i = 0$ *if* $\forall m : (i, m) \notin \mathcal{B}$.

Figure 2 depicts an example of how to apply Theorem 3.3 to obtain the optimal solution more efficiently. Theoretically, the key to show the optimality of (7) is via an explicit expression of the inverse of $\Theta$, i.e., $R = \Theta^{-1}$ using following theorem.

**Theorem 3.4.** *Given* $S, T \in \mathbb{S}^p$, *the bridge-block decomposition* $\mathcal{P}^{bbd} = \{\mathcal{V}_1, \ldots, \mathcal{V}_K\}$ *of* $\mathrm{supp}(T)$, *and a set of matrices* $\{\widehat{\Theta}_k \in \mathbb{S}_{++}^{|\mathcal{V}_k|}\}_{k=1}^K$, *the inverse of* $\Theta$ *in the form of* (7) *is derived as*

$$R_{ij} = \begin{cases} [\widehat{R}_k]_{\pi(i),\pi(j)} & \text{if } i, j \in \mathcal{V}_k, \\ T_{ij} & \text{if } (i,j) \in \mathcal{B}, \\ \sqrt{S_{ii}S_{jj}} \cdot g_{ij}(R) & \text{otherwise}, \end{cases} \tag{8}$$

*where* $\widehat{R}_k = [\widehat{\Theta}_k]^{-1}$ *and for each* $i, j$ *in different clusters,* $g_{ij}$ *is given as*

$$g_{ij}(R) = \prod_{t=0}^{2T} R_{u_t,u_{t+1}} / \sqrt{S_{u_t,u_t} S_{u_{t+1},u_{t+1}}}, \quad \text{where } u_0 \triangleq i, \ u_{2T+1} \triangleq j, \tag{9}$$

*in which* $T - 1$ *is the number of bridges in* $d_{ij}$, $g_{ij} = 0$ *if* $d_{ij} = \emptyset$, *and* $u$ *denotes a sequence of 'incident' bridges, i.e.,* $d_{ij} \cap \mathcal{B} = \{(u_1, u_2), \ldots, (u_{2T-1}, u_{2T})\}$ *following the orders of* $d_{ij}$ *while preserving the elements that are bridges.*

We note that all terms in (9) can be computed via (8) and Theorem 3.4 itself generally holds for arbitrary $S, T$, and $\{\widehat{\Theta}_k\}_{k=1}^K$. Detailed proofs and discussions are deferred to Appendix A.

Theorem 3.4 is theoretically critical as it provides an explicit form of the inverse of $\Theta$, which was previously difficult to compute. It is vital for demonstrating the optimality of $\Theta$ by connecting the KKT system of the original large-scale problem to the KKT systems of smaller sub-problems through the bridge-block decomposition. Consequently, together with MTP$_2$ properties, we show that $\Theta$ in the form of (7) satisfies the KKT condition of Problem (1) in Appendix B.

**The role of** MTP$_2$ **constraints:** Although Theorem 3.4 can be applied without requiring MTP$_2$ properties, it is important to highlight that these properties serve as sufficient conditions to demonstrate

the optimality of $\boldsymbol{\Theta} = \boldsymbol{R}^{-1}$. By incorporating the MTP$_2$ constraints, the non-smoothness in graphical lasso is eliminated, resulting in simplified optimality conditions as shown below:

$$
\begin{array}{llr}
\forall i: & -R_{ii} + S_{ii} = 0 & -R_{ii} + S_{ii} = 0 \quad \text{(10a)} \\
\forall \Theta_{ij} \neq 0: & -R_{ij} + S_{ij} + \lambda_{ij} \cdot \text{sign}(\Theta_{ij}) = 0 \quad \Longrightarrow & -R_{ij} + S_{ij} - \lambda_{ij} = 0 \quad \text{(10b)} \\
\forall \Theta_{ij} = 0: & |-R_{ij} + S_{ij}| \leq \lambda_{ij} & -R_{ij} + S_{ij} - \lambda_{ij} \leq 0 \quad \text{(10c)}
\end{array}
$$

Simultaneously, $\boldsymbol{R}$ becomes non-negative ($\geq \boldsymbol{0}$). As a result, The most challenging part of the KKT conditions $-R_{ij} + S_{ij} - \lambda_{ij} \leq 0$ holds under MTP$_2$ properties. In Appendix C, we will elaborate the details and explore additional conditions under which we can extend the applicability of Theorem 3.3 to the traditional graphical lasso.

**Proposed solving framework:** In practical applications, it is often more efficient to employ the bridge-block decomposition technique instead of directly manipulating full-sized matrices. This approach involves breaking down the problem into smaller isolated sub-problems. By addressing these sub-problems individually, we can compute the optimal solution by utilizing the solutions obtained from each sub-problem, as outlined in Theorem 3.3. This approach offers several advantages:

- **Significant reduction in computational cost**. Foremost, the cost of bridge-block decomposition is cheap. Suppose that we use a BCD solver of complexity $\mathcal{O}\left(p^4\right)$, then the total cost is reduced to $\sum_{k=1}^{K} \mathcal{O}(|\mathcal{V}_k|^4) \ll \mathcal{O}(p^4)$, where $\sum_k |\mathcal{V}_k| = p$. This can prompt an enormous difference.
- **Considerable reduction in memory cost**. The memory cost is typically troublesome for large-scale problems as each full-dimensional matrix contains $p^2$ elements. Theorem 3.3 can avoid generating a number of full-dimensional intermediate variables during computation.
- **Potential speed-up via parallel computing**. The sub-problems can be optimized independently, which allows parallel computing for significant speed-up.

### 3.3 Connection to Existing Research

From Theorem 3.3, we obtain a very interesting result that the $(i, j)$-th entries of $\boldsymbol{\Theta}$ admits an explicit solution, i.e., $\Theta_{ij} = -T_{ij} / \left(S_{ii}S_{jj} - T_{ij}^2\right)$ if edge $(i, j)$ is a bridge in the thresholded graph supp $(\boldsymbol{T})$. To the best of our knowledge, this result, shown in Corollary 3.5, is the first sufficient condition for an edge belonging to optimal graph supp $(\boldsymbol{\Theta}^\star)$.

**Corollary 3.5.** *If edge $(i, j)$ is a bridge in the thresholded graph supp $(\boldsymbol{T})$, then $(i, j)$ is also a bridge in the optimal graph supp $(\boldsymbol{\Theta}^\star)$.*

Corollary 3.5 can be obtained as follows. By the definition of a bridge, from nodes $i$ to $j$ the path $d_{ij} = \{(i, j)\}$ is the only path in the thresholded graph, and removing it would result in an increase in the number of components. Given that supp$(\boldsymbol{\Theta}^\star) \subseteq$ supp$(\boldsymbol{T})$ and $\Theta_{ij} \neq 0$, it follows that $d_{ij} = \{(i, j)\}$ remains the unique path from nodes $i$ to $j$ in the optimal graph. This indicates that the edge $(i, j)$ continues to function as a bridge.

In the literature, only some necessary conditions for edges in the thresholded graph supp $(\boldsymbol{T})$ to be retained in the optimal graph supp $(\boldsymbol{\Theta}^\star)$ have been established. For instance, [13] and [20] indicate that supp $(\boldsymbol{\Theta}^\star) \subseteq$ supp $(\boldsymbol{T})$, which implies that $(i, j) \notin$ supp $(\boldsymbol{\Theta}^\star)$ if $(i, j) \notin$ supp $(\boldsymbol{T})$.

Another explicit solution is also applicable when $k$-th cluster only contains one node, i.e., $\mathcal{V}_k = \{i\}$, then $\widehat{\boldsymbol{\Theta}}_k = 1/S_{ii}$. As a consequence, Theorem 3.3 generalizes to the following corollary for acyclic graph which admits a bridge-block decomposition into $p$ clusters.

**Corollary 3.6.** *Suppose that the thresholded graph supp $(\boldsymbol{T})$ is acyclic, then the optimal graph supp $(\boldsymbol{\Theta}^\star)$ is also acyclic and $\boldsymbol{\Theta}^\star$ admits a closed-form solution as*

$$
\Theta_{ij}^\star = \begin{cases}
\frac{1}{S_{ii}} \left(1 + \sum_{m \in \mathcal{N}(i)} \frac{T_{im}^2}{S_{ii}S_{mm} - T_{im}^2}\right) & \text{if } i = j, \\
-\frac{T_{ij}}{S_{ii}S_{jj} - T_{ij}^2} & \text{if } (i, j) \in \mathcal{B}, \\
0 & \text{otherwise.}
\end{cases} \quad \text{(11)}
$$

The proof directly follows Theorem 3.3. Corollary 3.6 coincides with [20, Theorem 3] and [38]. Compared to our results in Theorem 3.3, explicit solutions in literature heavily depend on the graph

structure. Hence, our theory appears as the first to reveal that, fundamentally, it is the edge property, i.e., whether the edge is a bridge, that decides the existence of explicit solutions.

In conclusion, our proposed methods generalize the existing results with more profound understandings, offering significant potential for solving large-scale precision matrix estimation problems.

## 4 Numerical Simulations

We conduct experiments on synthetic and real-world data to evaluate how the proposed method accelerates the convergence of existing algorithms compared to algorithms that do not exploit it. All experiments were conducted on 2.60GHZ Xeon Gold 6348 machines and Linux OS. All methods are implemented in MATLAB and the state-of-the-art methods we consider includes

- **BCD**: Block Coordinate Descent [19] of complexity $\mathcal{O}(p^4)$.
- **PGD**: Projected Gradient Descent [36] of complexity $\mathcal{O}(p^3)$.
- **PQN-LBFGS**: A projected quasi-Newton method with limited memory BFGS [51]. The complexity is $\mathcal{O}((m + p)p^2)$, where $m$ is the number of iterations stored for approximating the Hessian.
- **FPN**: Fast projected Newton-like method [37] of complexity $\mathcal{O}(p^3)$.

Note that all methods converge to the optimal solution. The target here is not to compare these methods but to evaluate how our proposed framework promotes the efficiency of these methods.

### 4.1 Synthetic Data Experiments

We use the processes described in [19] to generate the data. We begin with an underlying graph that has an adjacency matrix $\boldsymbol{A} \in \mathbb{S}^p$, and define $\boldsymbol{\Theta} = \delta \boldsymbol{I} - \boldsymbol{A}$, where $\delta = 1.05 \cdot \lambda_{\max}(\boldsymbol{A})$ and $\lambda_{\max}(\boldsymbol{A})$ represents the largest eigenvalue of $\boldsymbol{A}$. his ensures that $\boldsymbol{\Theta}$ is a positive definite matrix with off-diagonal elements being negative, making $\boldsymbol{\Theta}$ a randomly generated $M$-matrix. Next, we normalize $\boldsymbol{\Theta}$ by substituting it with $\boldsymbol{D\Theta D}$, where $\boldsymbol{D}$ is a suitably chosen diagonal matrix, ensuring that $\mathrm{diag}(\boldsymbol{\Theta}^{-1}) = \mathbf{1}$. We then sample $n = 10p$ data points from a Gaussian distribution $\mathcal{N}(\mathbf{0}, \boldsymbol{\Theta}^{-1})$ and calculate the sample covariance matrix as $\boldsymbol{S}$.

Following [25], we construct the regularization matrix $\boldsymbol{\Lambda}$. Briefly, given an initial estimate $\boldsymbol{\Theta}^{(0)}$, we set $\Lambda_{ij} = \chi/(\Theta_{ij}^{(0)} + \epsilon)$ when $i \neq j$ and $\Lambda_{ij} = 0$ when $i = j$. Here, $\chi > 0$ determines the sparsity level and $\varepsilon$ is a small positive constant, such as $10^{-3}$. Generally, a large penalty is expected on $\Theta_{ij}$ when $\Theta_{ij}^{(0)}$ is small. We efficiently compute $\boldsymbol{\Theta}^{(0)}$ using the following equation:

$$\Theta_{ij}^{(0)} = -T_{ij} \backslash \left( S_{ii}S_{jj} - T_{ij}^2 \right), \quad \forall i \neq j, \tag{12}$$

and then appropriately adjust the value of $\chi$ so that $\mathrm{supp}(\boldsymbol{T})$ has only one connected component, while $\mathrm{supp}(\boldsymbol{\Theta}^\star)$ can roughly recover the underlying structure.

We consider two scale-free random graphs as underlying with their typical structures in Figure 4.

**1. Barabasi-Albert (BA) graph** [52, 53] of order one. BA models generate random scale-free networks via preferential attachment, suitable for modeling various networks like the Internet, world wide web, protein interactions, citations, and social/online networks.

**2. Stochastic Block Model (SBM)** of networks [54], a.k.a. the community graph [55]. Stochastic block models serve as fundamental tools in network science, creating random networks based on community structures, where nodes within the same group are more likely to form connections.

In the experiments, we consider $p = 5000$ and $p = 20000$ and calculate the relative errors as follows:

$$\mathrm{RE}(\boldsymbol{\Theta}) = |f(\boldsymbol{\Theta}) - f(\boldsymbol{\Theta}^\star)| / |f(\boldsymbol{\Theta}^\star)|, \tag{13}$$

where we compare the relative errors at each iteration against the computational time. Here, $\boldsymbol{\Theta}^\star$ represents the optimal solution, and $f$ signifies the objective function of (1). For methods employing bridge-block decomposition (bbd), we first calculate $\widehat{\boldsymbol{\Theta}}_k^{(s)}$ for each cluster at the $s$-th iteration, followed by calculating the intermediate solution $\boldsymbol{\Theta}^{(s)}$ using (7). The computational time at the $s$-th iteration is then the sum of the costs to obtain $\mathcal{P}^{\mathrm{bbd}}$, calculate all $\widehat{\boldsymbol{\Theta}}_k^{(s)}$, and compute $\boldsymbol{\Theta}^{(s)}$ via (7).

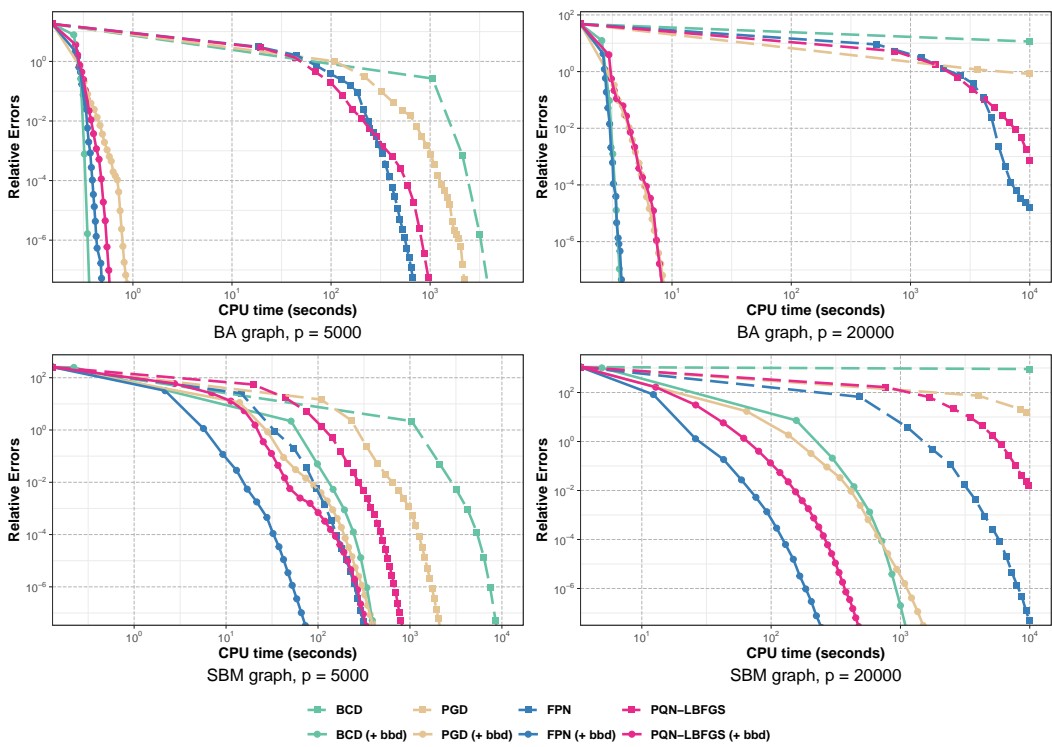

Figure 3: Relative errors of the objective values versus the computational time. Colors are used for distinguishing the state-of-the-art methods. Solid circle symbols stand for methods accelerated by bridge-block decomposition (bbd) and square symbols stand for methods without acceleration.

The results for $p = 5000$ and $p = 20000$ are displayed in Figure 3, with all results averaged over five realizations. We highlight that the extra computational cost compared to methods without acceleration is negligible as shown in Table 1. From the figures, we observe that:

- **Significant Speed-up:** Our proposed framework substantially accelerates state-of-the-art methods by one to four orders of magnitude.

- **Solving Otherwise Infeasible Problems:** When $p = 20000$, existing methods cannot optimize the problem within $10^4$ seconds. In contrast, our framework greatly speeds up convergence, making it possible to solve large-scale problems.

- **Higher Speed-up for Sparser Graph:** Proposed method achieves greater acceleration on the BA graph than the SBM graph, as the former exhibits a stronger sparsity pattern.

- **Higher Speed-up for Methods of Higher-complexity:** Figure 3 demonstrates that our method has a more significant impact on improving the efficiency of the BCD method, which has higher computational complexity and thus benefits more from dimension reduction.

To further shed light on the factors that determine the magnitudes of improvement, in the next sub-section, we conduct additional experiments that dive into how bridge-block decomposition boosts the performance of existing methods subject to different structures of the thresholded graph.

Table 1: Average computational time (seconds) of extra cost.

| | Conducting Bridge-block decomposition | | Computing $\boldsymbol{\Theta}$ from $\{\widehat{\boldsymbol{\Theta}}_k\}_{k=1}^{K}$ | |
| --- | --- | --- | --- | --- |
| | $p = 5k$ | $p = 20k$ | $p = 5k$ | $p = 20k$ |
| BA | 0.2 | 2.2 | 0.0 | 0.0 |
| SBM | 0.3 | 2.3 | 0.1 | 0.2 |

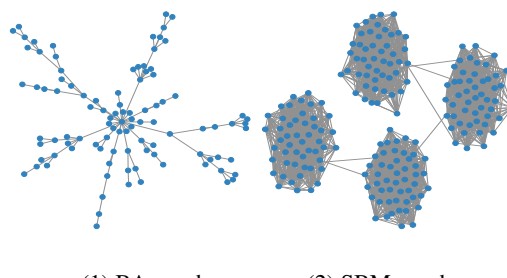
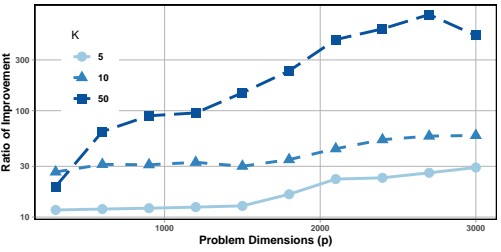

|  |  |  |
| :---: | :---: | :---: |
| (1) BA graph | (2) SBM graph |  |

Figure 4: Structures of BA graph and SBM graph.

Figure 5: Ratios of improvements for BCD method with respect to computational time.

## 4.2 Further Experiments on Ratios of Improvement

We investigate a scenario where the underlying graph is a community graph with a block-tridiagonal adjacency matrix, and each cluster contains an equal number of nodes ($p_1 = |\mathcal{V}_k|, \forall k$). Without loss of generality, we assume that the internal edges within clusters form a cycle. To generate $S$ and $\Lambda$, we follow the procedures in Section 4.1. We then modify $\Lambda$ to $\alpha \left( \mathbf{1}\mathbf{1}^T - A \right) + (1 - \alpha) \Lambda$, ensuring that $\text{supp} \left( T \right) = \text{supp} \left( A \right)$, where $\alpha \in [0, 1]$ and $A$ is the adjacency matrix of the considered graph.

We experiment with various values of $K$ (the number of clusters) and $p_1$ (the number of nodes in each cluster). For each configuration, we calculate the ratio of improvement, defined as the quotient of the time required to achieve $\text{RE} \left( \boldsymbol{\Theta} \right) < 10^{-6}$ without using bridge-block decomposition to the time when applying the decomposition. We conduct multiple trials of the BCD method and display the results in Figure 5. The results in Figure 5 suggest that the number of sub-graphs $K$ is a primary factor influencing the ratio values. The improvement is deemed tremendous, especially for large-scale settings. The involvement of bridge-block decomposition enables solving an MTP$_2$ GGMs with hundreds of millions of variables as long as we can conduce bridge-block decomposition. Consequently, many previously unattainable real-life applications can now be optimized.

## 4.3 Real-word Data Experiment

We consider learning the MTP$_2$ GGM for the Crop Image dataset available from the UCR Time Series Archive [56]. This dataset comprises $p = 24000$ pixels, where each pixel corresponds to a time series record capturing spectral information in $n = 46$ instances. These instances represent geometrically and radiometrically corrected satellite images. By examining this dataset, we can observe the temporal changes in the observed areas through the recorded measurements. To facilitate our analysis, the data has been pre-processed by [57] using an indexing technique. This preprocessing step enables us to work with compressed informative indexes instead of handling large image files.

Our goal is to perform graph-based clustering to the time series data that contains 46 observations, denoted as $\{\boldsymbol{y}_1, \ldots, \boldsymbol{y}_{46}\}$ using MTP$_2$ GGMs, where $\boldsymbol{y}_i \in \mathbb{R}^{24000}$. The data includes 24 classes corresponding to different indexed land covers, such as corn, barley, or lake. It is important to note that these labels are not involved in the clustering process but help evaluate the quality of the final graph-based clusters by computing the graph modularity [58]. Via analyzing the data, in Appendix D, we discuss that why learning MTP$_2$ GGMs is statistically meaningful in the context of clustering and show that the MTP$_2$ assumptions approximately hold for this dataset.

The initial estimate $\boldsymbol{\Theta}^{(0)}$ is computed using the method proposed in [59]. The regularization matrix $\boldsymbol{\Lambda}$ is determined using the approach in Section 4.1 with $\epsilon = 0.01$ and $\chi = 0.2$. We then compute the sample correlation matrix as $S \in \mathbb{S}^{24000}$. This results in a precision matrix estimation problem involving $5.76 \times 10^8$ variables to optimize, which is quite challenging for state-of-the-art methods without bridge-block decomposition.

Figure 6b shows the results of empirical convergence, and Figure 6a visualizes a local structure of $\text{supp} \left( \boldsymbol{\Theta} \right)^{\star}$ (modularity = 0.6849). In line with previous findings, our suggested approach greatly speeds up the convergence of all current algorithms by a minimum of three orders of magnitude. This demonstrates immense potential for managing large-scale sparse graphical models.

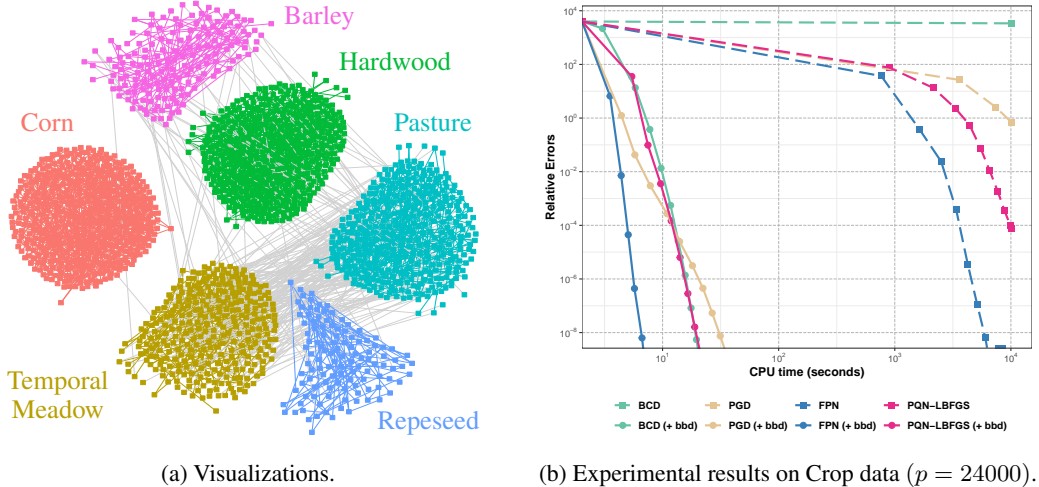

(a) Visualizations.

(b) Experimental results on Crop data ($p = 24000$).

Figure 6: Visualizations and experimental results of sparse MTP$_2$ GGMs for Crop data set. (a) A local structure of optimal graph with 2008 nodes. Nodes with matching labels are assigned the same color and connected by a matching edge color, while different groups of nodes are connected by gray edges. (b) Results of convergence to learn sparse MTP$_2$ GGMs for Crop data set.

While our paper's primary focus is not to uncover insights into the estimated MTP$_2$ GGMs for a deeper understanding of the underlying data, interesting phenomena can still be observed in the structures presented in Figure 6a. Notably, we find that the majority of edges exist within the same crop type, while connections between nodes associated with different crops are relatively sparse. This observation is valuable for clustering processes and aligns with our expectations, as stronger positive dependencies are often observed within the same class, while dependencies between different classes tend to be weaker.

Moreover, our graphical representation reveals more nuanced patterns. For instance, in Figure 6a of our manuscript, we observe a dense network of edges between two crop types, 'temporary meadow' and 'pasture', indicating a significantly stronger conditional dependency compared to other categories. This observation further supports our understanding. Hence, the inferred conditional dependency structure encapsulates the inherent interrelationships among different crops. As a result, our proposed framework based on bridge-block decomposition effectively facilitates learning in large-scale MTP$_2$ graphical models.

## 5   Conclusions and Discussions

Real-world sparse Gaussian graphical models often comprise subsets of variables that are densely connected to one another, while variables in different clusters maintain weak connections. However, standard estimation algorithms do not account for this property. In this paper, by introducing the concept of bridge, we leverage these characteristics to reveal an interesting finding: the optimal solution for Problem (1) equivalently admits a decomposed form via the bridge-block decomposition of the thresholded graph. We provide theoretical insights into the separability of optimal solutions and the existence of explicit expressions based on edge properties, specifically whether an edge is a bridge. This method surpasses conventional approaches that depend on unique graph structures, such as determining if a graph is acyclic. From the practical aspect, our proposed method offers a handy architect for accelerating any existing algorithms by handling a large-scale learning problem via a number of small and tractable sub-problems. Although our method is mainly developed for sparse large-scale graphs and therefore seems inapplicable for dense graph, as elaborated in the appendix C, it can also provide a quick way to obtain a solution, which can serve as either a starting point (warm-start) for numerical algorithms or an alternative for acquiring approximate solutions. Overall, our simple and provable approach demonstrates exceptional performance and paves a novel way for more extensive designs of large-scale MTP$_2$ Gaussian graphical models.

## 6 Acknowledgements

This work was supported by the Hong Kong Research Grants Council GRF 16207820, 16310620, and 16306821, the Hong Kong Innovation and Technology Fund (ITF) MHP/009/20, and the Project of Hetao Shenzhen-Hong Kong Science and Technology Innovation Cooperation Zone under Grant HZQB-KCZYB-2020083. We would also like to thank the anonymous reviewers for their valuable feedback on the manuscript.

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

## Appendix

In what follows, we present technical proofs omitted in the main content. In Section A, we introduce important properties of matrix $\boldsymbol{R}$ and prove Theorem 3.4. In Section B, we prove the main result, i.e., Theorem 3.3. In Section C, we introduce some extension of the proposed framework to broaden its applicability. In Section D, we will comment on real-world experiments and explain why MTP$_2$ assumption approximately holds for Crop Image dataset.

## A   Proof of Theorem 3.4

This section contains a detailed description of how we construct the matrix $\boldsymbol{R}$ and prove Theorem 3.4 based on the properties of $\boldsymbol{R}$. The proof relies on the concept of a 'generalized path product', which involves a set of edges labeled as a 'bridge path'. The matrix $\boldsymbol{R}$ has several notable properties that are essential in demonstrating Theorem 3.3.

### A.1   Definitions and Important Properties

To begin, let us define the 'bridge path' $\mathcal{B}_{ij}$ for nodes $i$ and $j$ that belong to different clusters, i.e., $\psi(i) \neq \psi(j)$ based on the bridge set $\mathcal{B}$ in the thresholded graph $G$. The bridge path $\mathcal{B}_{ij}$ is defined as the set of bridges in any path $d_{ij}$ from node $i$ to node $j$, where each element in $\mathcal{B}_{ij}$ is a bridge in $\mathcal{B}$ and the elements in $\mathcal{B}_{ij}$ follow the order of $d_{ij}$ while preserving the elements in $\mathcal{B}$. Symbolically, we can express this as:

$$\mathcal{B}_{ij} = d_{ij} \cap \mathcal{B} = \{(u_1, u_2), (u_3, u_4), \ldots, (u_{2T-1}, u_{2T})\}, \tag{14}$$

where $T$ denotes the number of clusters in any path from nodes $i$ to $j$. It is important to note the following:

- Each element in $\mathcal{B}_{ij}$ is a bridge in the thresholded graph.
- The elements in $\mathcal{B}_{ij}$ follow the orders of $d_{ij}$. This means that for any path from nodes $i$ to $j$, the path will visit $(u_{2i-1}, u_{2i})$, then visit $(u_{2j-1}, u_{2j})$ whenever we have $i < j$. consequently, we have

$$\psi(i) = \psi(u_1), \quad \psi(u_2) = \psi(u_3), \ldots, \psi(u_{2T-2}) = \psi(u_{2T-1}), \quad \psi(u_{2T}) = \psi(j). \tag{15}$$

- For any path $d_{ij}$, the bridge path $\mathcal{B}_{ij} = d_{ij} \cap \mathcal{B}$ is unique.
- $\mathcal{B}_{ij}$ is defined as an empty set if there is no path from nodes $i$ to $j$.

It is important to mention that if the thresholded graph is acyclic, the bridge path can be reduced to a traditional path. In this scenario, all edges in the path $d_{ij}$ become bridges, and the bridge path $\mathcal{B}_{ij}$ is equivalent to the path $d_{ij}$, which can be expressed as $\mathcal{B}_{ij} = d_{ij}$.

We define the matrix $\boldsymbol{R} \in \mathbb{S}^p$ based on the definition of bridge-path, as shown below:

$$R_{ij} = \begin{cases} [\widehat{\boldsymbol{R}}_k]_{\pi(i),\pi(j)} & \text{if } i,j \in \mathcal{V}_k, \\ T_{ij} & \text{if } (i,j) \in \mathcal{B}, \\ \sqrt{S_{ii}S_{jj}} \cdot g_{ij}(\boldsymbol{R}) & \text{otherwise}, \end{cases} \tag{16}$$

where $\widehat{\boldsymbol{R}}_k = [\widehat{\boldsymbol{\Theta}}_k]^{-1}$ and for each $i, j$ in different clusters, $g_{ij}$ is given as

$$g_{ij}(\boldsymbol{R}) = \prod_{t=0}^{2T} R_{u_t,u_{t+1}} \Big/ \sqrt{S_{u_t,u_t} S_{u_{t+1},u_{t+1}}}, \quad \text{where } u_0 \triangleq i, \ u_{2T+1} \triangleq j, \tag{17}$$

where $u$ denotes the bridge path $\mathcal{B}_{ij}$ and $g_{ij}(\boldsymbol{R}) = 0$ if $\mathcal{B}_{ij} = \emptyset$.

*Remark* A.1.  Clearly, we have $R_{ij} = R_{ji}$ for any $i$ and $j$.

**Example**: For instance, using Figure 7 as an example, we consider a path from node 1 to 10 as

$$d_{1,10} = \{(1,4), (4,5), \underbrace{(5,6)}_{\text{bridge}}, (6,7), (7,8), (8,9), \underbrace{(9,10)}_{\text{bridge}}\}, \tag{18}$$

the bridge path is then given as

$$\mathcal{B}_{1,10} = \{(5,6), (9,10)\}, \tag{19}$$

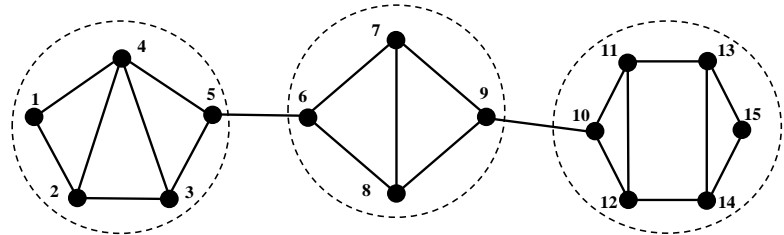

Figure 7: An example of computing bridge path. For example, from node 1 to 10, all the paths will cover the bridges $(5,6)$ and $(9,10)$. Hence, we have $\mathcal{B}_{1,10} = \{(5,6),(9,10)\}$.

and $R_{1,10}$ is computed as

$$
\begin{aligned}
R_{1,10} &= \sqrt{S_{1,1}} \cdot g_{1,10}\left(\boldsymbol{R}\right) \cdot \sqrt{S_{10,10}} \\
&= \sqrt{S_{1,1}} \cdot \frac{R_{1,5}}{\sqrt{S_{1,1}S_{5,5}}} \cdot \frac{R_{5,6}}{\sqrt{S_{5,5}S_{6,6}}} \cdot \frac{R_{6,9}}{\sqrt{S_{6,6}S_{9,9}}} \cdot \frac{R_{9,10}}{\sqrt{S_{9,9}S_{10,10}}} \cdot \frac{R_{10,10}}{\sqrt{S_{10,10}S_{10,10}}} \cdot \sqrt{S_{10,10}}.
\end{aligned}
\tag{20}
$$

Particularly, we note that

- The bridge path $\mathcal{B}_{1,10}$ can not be written as $\{(9,10),(5,6)\}$ as $d_{1,10}$ would always visit the bridge $(5,6)$ before visiting $(9,10)$.
- The bridge path $\mathcal{B}_{1,10}$ can not be written as $\{(6,5),(9,10)\}$ as $d_{1,10}$ leave first cluster at node 5, then enter second cluster at node 6. Therefore, we have $\psi(1) = \psi(5)$.
- It is easy to observe that $\mathcal{B}_{10,1} = \{(10,9),(6,5)\}$. Based on definitions, we can obtain $R_{10,1} = R_{1,10}$.
- Specially, if the matrix $\boldsymbol{S}$ is given as a correlation matrix, i.e., $\mathrm{diag}(\boldsymbol{S}) = \mathbf{1}$, then $R_{1,10} = R_{1,5} \cdot R_{5,6} \cdot R_{6,9} \cdot R_{9,10} \cdot R_{10,10}$, which can be seen as a continued produce on a path, known as path product.

The matrix $\boldsymbol{R}$ shares some important properties introduced as follows.

**Lemma A.2.** *For any $i,j \in \mathcal{V}_k$, $\widehat{\boldsymbol{R}}_k$ together with $\widehat{\boldsymbol{\Theta}}_k$ would satisfy the following equations*

$$
-[\widehat{\boldsymbol{R}}_k]_{\pi(i),\pi(j)} + S_{i,j} = 0 \qquad \text{if } i = j, \tag{21a}
$$

$$
-[\widehat{\boldsymbol{R}}_k]_{\pi(i),\pi(j)} + S_{i,j} - \Lambda_{i,j} = 0 \qquad \text{if } i \neq j, [\widehat{\boldsymbol{\Theta}}_k]_{\pi(i),\pi(j)} \neq 0, \tag{21b}
$$

$$
-[\widehat{\boldsymbol{R}}_k]_{\pi(i),\pi(j)} + S_{i,j} - \Lambda_{i,j} \leq 0 \qquad \text{if } i \neq j, [\widehat{\boldsymbol{\Theta}}_k]_{\pi(i),\pi(j)} = 0, \tag{21c}
$$

*Proof.* Let's denote $\Gamma_{\pi(i),\pi(j)}$ as the dual variables associated with the constraints $[\widehat{\boldsymbol{\Theta}}_k]_{\pi(i),\pi(j)} \leq 0$. With $\widehat{\boldsymbol{R}}_k = \widehat{\boldsymbol{\Theta}}_k^{-1}$, the KKT conditions include

$$
\text{stationarity:} \qquad -\widehat{\boldsymbol{R}}_k + \boldsymbol{S}_{\mathcal{V}_k} - \boldsymbol{\Lambda}_{\mathcal{V}_k} + \boldsymbol{\Gamma}_{\mathcal{V}_k} = \mathbf{0}, \tag{22a}
$$

$$
\text{primal feasibility:} \qquad [\widehat{\boldsymbol{\Theta}}_k]_{\pi(i),\pi(j)} \leq 0, \ \forall \pi(i) \neq \pi(j) \tag{22b}
$$

$$
\text{dual feasibility:} \qquad \Gamma_{\pi(i),\pi(j)} \geq 0, \ \forall \pi(i) \neq \pi(j) \tag{22c}
$$

$$
\text{complementary slackness:} \qquad [\widehat{\boldsymbol{\Theta}}_k]_{\pi(i),\pi(j)} \cdot \Gamma_{\pi(i),\pi(j)} = 0, \ \forall \pi(i) \neq \pi(j) \tag{22d}
$$

We can eliminate the dual variables as follows:

- For $[\widehat{\boldsymbol{\Theta}}_k]_{\pi(i),\pi(j)} < 0$, the complementary slackness leads to $\Gamma_{\pi(i),\pi(j)} = 0$, which implies $-[\widehat{\boldsymbol{R}}_k]_{\pi(i),\pi(j)} + S_{i,j} - \Lambda_{i,j} = 0$.

- For $[\widehat{\boldsymbol{\Theta}}_k]_{\pi(i),\pi(j)} = 0$, we have $\Gamma_{\pi(i),\pi(j)} \geq 0$, which implies $-[\widehat{\boldsymbol{R}}_k]_{\pi(i),\pi(j)} + S_{i,j} - \Lambda_{i,j} \leq 0$.

Hence, Lemma A.2 is obtained. $\qquad \square$

In particular, we note that $[\widehat{\boldsymbol{R}}_k]_{\pi(i),\pi(i)} = S_{i,i}$ holds for any $i \in \mathcal{V}_k$ according to (21a). As a result, the following property can be easily verified via the definitions.

**Corollary A.3.** *Suppose that two nodes $i$ and $j$ are in different clusters, i.e., $\psi(i) \neq \psi(j)$, and the bridge path is not empty and denoted by $\mathcal{B}_{ij}$ with form of (14), then we have*

$$R_{ij} = \frac{R_{i,u_t} \cdot R_{u_t,j}}{S_{u_t,u_t}}, \qquad \forall t = 0,\dots,2T+1, \tag{23}$$

*Proof.* Corollary A.3 can be easily verified via definitions. If $t = 0$ or $t = 2T + 1$, Corollary A.3 holds as $R_{i,i} = S_{i,i}$ and $R_{j,j} = S_{j,j}$. Let $s$ be a number that satisfies $0 \leq s \leq 2T$. For any $(u_s, u_{s+1}) \in \mathcal{B}_{ij}$, we have

$$R_{i,j} \cdot S_{u_s,u_s} = \underbrace{\sqrt{S_{i,i}} \cdot \frac{R_{u_0,u_1}}{\sqrt{S_{u_0,u_0}S_{u_1,u_1}}} \cdots \cdots \frac{R_{u_{s-1},u_s}}{\sqrt{S_{u_{s-1},u_{s-1}}S_{u_s,u_s}}} \cdot \sqrt{S_{u_s,u_s}}}_{R_{i,u_s}}$$

$$\underbrace{\cdot \sqrt{S_{u_s,u_s}} \cdot \frac{R_{u_s,u_{s+1}}}{\sqrt{S_{u_s,u_s}S_{u_{s+1},u_{s+1}}}} \cdots \cdots \frac{R_{u_{2T-1},u_{2T}}}{\sqrt{S_{u_{2T-1},u_{2T-1}}S_{u_{2T},u_{2T}}}} \cdot \sqrt{S_{j,j}}}_{R_{u_s,j}} \cdot \tag{24}$$

Clearly, Corollary A.3 is obtained. $\qquad\square$

The aforementioned properties of matrix $\boldsymbol{R}$ are important for establishing the proof of Theorem 3.4 and Theorem 3.3.

## A.2 Proof of Theorem 3.4

*Proof.* Let $\boldsymbol{F} = \boldsymbol{\Theta R}$, where $\boldsymbol{\Theta}$ and $\boldsymbol{R}$ have the forms of (7) and (8), respectively. The target is to show $F_{ii} = 1, \forall i$ and $F_{ij} = 0, \forall i \neq j$ using the following results we have derived.

**(a)** For any $i, j \in \mathcal{V}_k$, we have

$$\boldsymbol{\Theta}_{ij} = \begin{cases} [\widehat{\boldsymbol{\Theta}}_k]_{\pi(i),\pi(j)} + \frac{1}{S_{ii}} \cdot \sum_{(i,m)\in\mathcal{B}} \frac{T_{im}^2}{S_{ii}S_{mm}-T_{im}^2} & \text{if } i = j, \\ [\widehat{\boldsymbol{\Theta}}_k]_{\pi(i),\pi(j)} & \text{if } i \neq j, \end{cases} \tag{25}$$

and $R_{ij} = [\widehat{\boldsymbol{R}}_k]_{\pi(i),\pi(j)}$ according to the definitions.

**(b)** For any nodes $i$ and $j$ in different clusters, suppose that $\mathcal{B}_{ij}$ is not empty and edge $(u, v)$ is a bridge in $\mathcal{B}_{ij}$, then $R_{ij}S_{uu} = R_{iu}R_{uj}$ and $R_{ij}S_{vv} = R_{iv}R_{vj}$ hold according to Corollary A.3.

**(c)** The matrix $\widehat{\boldsymbol{F}}_k$, which is defined as $\widehat{\boldsymbol{F}}_k = \widehat{\boldsymbol{R}}_k\widehat{\boldsymbol{\Theta}}_k$ is equal to the identity matrix, i.e., $\widehat{\boldsymbol{F}}_k = \boldsymbol{I}$. In other words, we have

$$[\widehat{\boldsymbol{F}}_k]_{\pi(i),\pi(i)} = 1, \text{ and } [\widehat{\boldsymbol{F}}_k]_{\pi(i),\pi(j)} = 0, \quad \forall i \neq j \text{ and } i, j \in \mathcal{V}_k. \tag{26}$$

Based on (a), (b), and (c), we first show that $F_{ii} = 1$ holds for any $i \in \{1, \dots, p\}$. We suppose that node $i$ is in $k$-the cluster, i.e., $k = \psi(i)$. Then, by using $\forall m \notin (\mathcal{V}_k \cup \mathcal{N}(i)) : \Theta_{im} = 0$ from definitions in (7), we have

$$F_{ii} = \sum_{m=1}^p R_{im}\Theta_{im} = \sum_{m\in\mathcal{V}_k\setminus\{i\}} R_{im}\Theta_{im} + R_{ii}\Theta_{ii} + \sum_{(i,m)\in\mathcal{B}} R_{im}\Theta_{im}$$

$$\stackrel{(a)}{=} \sum_{m\in\mathcal{V}_k\setminus\{i\}} [\widehat{\boldsymbol{R}}_k]_{\pi(i),\pi(m)}[\widehat{\boldsymbol{\Theta}}_k]_{\pi(i),\pi(m)} + [\widehat{\boldsymbol{R}}_k]_{\pi(i),\pi(i)}[\widehat{\boldsymbol{\Theta}}_k]_{\pi(i),\pi(i)}$$

$$+ S_{ii} \cdot \frac{1}{S_{ii}} \sum_{(i,m)\in\mathcal{B}} \frac{T_{im}^2}{S_{ii}S_{mm}-T_{im}^2} + \sum_{(i,m)\in\mathcal{B}} T_{im}\left(-\frac{T_{im}}{S_{ii}S_{mm}-T_{im}^2}\right)$$

$$= \sum_{m\in\mathcal{V}_k\setminus\{i\}} [\widehat{\boldsymbol{R}}_k]_{\pi(i),\pi(m)}[\widehat{\boldsymbol{\Theta}}_k]_{\pi(i),\pi(m)} + [\widehat{\boldsymbol{R}}_k]_{\pi(i),\pi(i)}[\widehat{\boldsymbol{\Theta}}_k]_{\pi(i),\pi(i)} \stackrel{(c)}{=} [\widehat{\boldsymbol{F}}_k]_{\pi_k(i),\pi_k(i)} = 1. \tag{27}$$

Therefore, $T_{ii} = 1$ holds for any $i \in \{1, \ldots, p\}$.

Let us proceed with the proof that $F_{ij} = \sum_{m=1}^{p} R_{im}\Theta_{mj} = 0$ for any $i \neq j$. Without loss of generality, we suppose that node $j$ is in the $k$-th cluster, i.e., $k = \psi(j)$. According to the definitions, we have

$$\forall m \notin (\mathcal{V}_k \cup \mathcal{N}(j)) : \Theta_{mj} = 0. \tag{28}$$

Here, there would be three cases to analyze.

The first case happens where $i \in \mathcal{V}_k$, which means that nodes $i$ and $j$ are in the same cluster. In this case, we have

$$
\begin{aligned}
F_{ij} &= \sum_{m=1}^{p} R_{im}\Theta_{mj} = \sum_{m \in \mathcal{V}_k \setminus \{j\}} R_{im}\Theta_{mj} + R_{ij}\Theta_{jj} + \sum_{(m,j) \in \mathcal{B}} R_{im}\Theta_{mj} \\
&\overset{(a)}{=} \sum_{m \in \mathcal{V}_k \setminus \{j\}} [\widehat{\boldsymbol{R}}_k]_{\pi(i),\pi(m)} [\widehat{\boldsymbol{\Theta}}_k]_{\pi(m),\pi(j)} \\
&\quad + [\widehat{\boldsymbol{R}}_k]_{\pi(i),\pi(j)} \left( [\widehat{\boldsymbol{\Theta}}_k]_{\pi(j),\pi(j)} + \frac{1}{S_{jj}} \sum_{(m,j) \in \mathcal{B}} \frac{T_{mj}^2}{S_{mm}S_{jj} - T_{mj}^2} \right) + \sum_{(m,j) \in \mathcal{B}} R_{im}\Theta_{mj} \\
&\overset{(c)}{=} [\widehat{\boldsymbol{F}}_k]_{\pi_k(i),\pi_k(j)} + R_{ij} \left( \frac{1}{S_{jj}} \sum_{(m,j) \in \mathcal{B}} \frac{T_{mj}^2}{S_{mm}S_{jj} - T_{mj}^2} \right) + \sum_{(m,j) \in \mathcal{B}} R_{im}\Theta_{mj} \\
&\overset{(b,c)}{=} 0 + R_{ij} \left( \frac{1}{S_{jj}} \sum_{(m,j) \in \mathcal{B}} \frac{T_{mj}^2}{S_{mm}S_{jj} - T_{mj}^2} \right) + \sum_{(m,j) \in \mathcal{B}} \frac{R_{ij}R_{jm}}{S_{jj}} \left( -\frac{T_{mj}}{S_{mm}S_{jj} - T_{mj}^2} \right) \\
&= 0. \tag{29}
\end{aligned}
$$

The second case happens where $i \notin \mathcal{V}_k$, but there is no path between nodes $i$ and $j$. In such a case, for any node $m$ in $\mathcal{V}_k \cup \mathcal{N}(j)$, there is also no path from nodes $i$ to $m$. Consequently, based on the definition of $\boldsymbol{R}$, we have

$$\forall m \in \mathcal{V}_k \cup \mathcal{N}(j) : \quad R_{im} = 0. \tag{30}$$

As a result, we obtain

$$F_{ij} = \sum_{m=1}^{p} R_{im}\Theta_{mj} = \sum_{m \in \mathcal{V}_k \cup \mathcal{N}(j)} R_{im}\Theta_{mj} = 0. \tag{31}$$

The third case happens where $i \notin \mathcal{V}_k$ and there exists a path from nodes $i$ to $j$. Then we have $\psi(i) \neq \psi(j)$ and the bridge path $\mathcal{B}_{ij}$ is denoted as

$$\mathcal{B}_{ij} = \{(u_1, u_2), (u_3, u_4), \ldots, (u_{2T-1}, u_{2T})\}, \tag{32}$$

where $\psi(u_1) = \psi(i)$ and $\psi(u_{2T}) = \psi(j) = k$. In particular, we emphasize that $u_{2T} \in \mathcal{V}_k$ as $(u_{2T-1}, u_{2T})$ is a bridge to enter the cluster that contains $j$. Therefore, we have

$$F_{ij} = \sum_{m=1}^{p} R_{im}\Theta_{mj} \overset{(b)}{=} \frac{R_{i,u_{2T}}}{S_{u_{2T},u_{2T}}} \cdot \sum_{m=1}^{p} R_{u_{2T},m}\Theta_{mj} = 0, \tag{33}$$

where the last equality is from (29) by using $u_{2T} \in \mathcal{V}_k$. In conclusion, via enumerating all the possible cases, we have $\boldsymbol{F} = \boldsymbol{I}$, which means that $\boldsymbol{R}\boldsymbol{\Theta} = \boldsymbol{I}$. $\qquad \square$

*Remark* A.4. Theorem 3.4 generally holds for any $\boldsymbol{T}$, $\boldsymbol{S}$ and $\{\widehat{\boldsymbol{\Theta}}_k \in \mathbb{S}_{++}^{|\mathcal{V}_k|}\}_{k=1}^{K}$. It does not utilize MTP$_2$ properties from $\boldsymbol{\Theta}$. Hence, Theorem 3.4 itself may be extended to graphical models beyond MTP$_2$ assumption.

# B  Proof of Main Results

To prove Theorem 3.3, we require some preliminaries on $\text{MTP}_2$ properties.

**Lemma B.1.** *[11, Theorem 1] A real symmetric matrix $\boldsymbol{X} \in \mathbb{S}^p$, which admits the form:*

$$\forall i \neq j: \quad X_{ij} \leq 0, \tag{34}$$

*is a non-singular $M$-matrix if and only if one of the following statements holds: (1) Every eigenvalue of $\boldsymbol{X}$ is positive; (2) $\boldsymbol{X}$ is inverse-positive. That is, $\boldsymbol{X}^{-1}$ exists and $\boldsymbol{X}^{-1} \geq \boldsymbol{0}$.*

A direct outcome of Lemma B.1 is that the matrix $\boldsymbol{R}$ is a non-negative matrix.

**Corollary B.2.** *The matrix $\boldsymbol{R}$ given by (8) satisfies $\boldsymbol{R} \geq \boldsymbol{0}$.*

Corollary B.2 holds according to the fact that

$$\forall k \in \{1, \ldots, K\}: \quad \widehat{\boldsymbol{\Theta}}_k \in \mathcal{M}^{p_k} \quad \Rightarrow \quad \widehat{\boldsymbol{R}_k} \geq \boldsymbol{0}. \tag{35}$$

Now, we begin proving Theorem 3.3.

**Proof of Theorem 3.3:** First of all, we show that the matrix $\boldsymbol{\Theta}$, given as

$$\boldsymbol{\Theta}_{i,j} = \begin{cases} [\widehat{\boldsymbol{\Theta}}_k]_{\pi(i),\pi(i)} + \zeta_i & \text{if } i = j \in \mathcal{V}_k, \\ [\widehat{\boldsymbol{\Theta}}_k]_{\pi(i),\pi(j)} & \text{if } i \neq j \text{ and } i, j \in \mathcal{V}_k, \\ -T_{ij}/(S_{ii}S_{jj} - T_{ij}^2) & \text{if } (i,j) \in \mathcal{B}, \\ 0 & \text{otherwise.} \end{cases} \tag{36}$$

is primal feasible and positive definite. Based on Assumption 2.1, which states that $S_{ij} < \sqrt{S_{ii}S_{jj}}$, we know that $T_{ij}^2 \leq (S_{ij} - \Lambda_{ij})^2 \leq S_{ij}^2 < S_{ii}S_{jj}$. Hence, for any $i \neq j$, we have $\Theta_{ij} \leq 0$ due to $-T_{ij}/(S_{ii}S_{jj} - T_{ij}^2) \leq 0$ and the primal feasibility of each $\widehat{\boldsymbol{\Theta}}_k$. Additionally, since we have $\boldsymbol{R} \geq \boldsymbol{0}$ according to Corollary B.2, we can conclude that $\boldsymbol{\Theta}$ is inverse-positive according to Theorem 3.4, which implies that it is a positive-definite matrix. In other words, $\boldsymbol{\Theta}$ is both primal feasible and inverse-positive, and we can express this as $\boldsymbol{\Theta} \succ \boldsymbol{0}$ based on Lemma B.1.

In second step, we show that $\boldsymbol{\Theta} = \boldsymbol{R}^{-1}$ satisfies the optimality condition listed as follows, i.e.,

$$-\left[\boldsymbol{\Theta}^{-1}\right]_{ij} + S_{ij} = 0 \qquad \text{if } i = j, \tag{37a}$$

$$-\left[\boldsymbol{\Theta}^{-1}\right]_{ij} + S_{ij} - \Lambda_{ij} = 0 \qquad \text{if } i \neq j, \Theta_{ij} \neq 0, \tag{37b}$$

$$-\left[\boldsymbol{\Theta}^{-1}\right]_{ij} + S_{ij} - \Lambda_{ij} \leq 0 \qquad \text{if } i = j, \Theta_{ij} = 0. \tag{37c}$$

For (37a), we let $k = \psi(i)$. Then, according to (21a) in Lemma A.2, we have

$$-[\boldsymbol{\Theta}^{-1}]_{ij} + S_{ij} = -[\widehat{\boldsymbol{R}}_k]_{\pi(i),\pi(i)} + S_{ij} = 0. \tag{38}$$

For (37b), as $\Theta_{ij} \neq 0$, the pair $(i,j)$ either satisfies $(i,j) \in \mathcal{B}$ or $\psi(i) = \psi(j) \triangleq k$. If $(i,j) \in \mathcal{B}$, then by using $R_{ij} = T_{ij}$, we have

$$-[\boldsymbol{\Theta}^{-1}]_{ij} + S_{ij} - \Lambda_{ij} = -R_{ij} + T_{ij} = 0. \tag{39}$$

If nodes $i$ and $j$ are in the same cluster, then according to (21b) in Lemma A.2, we have $\Theta_{ij} = [\widehat{\boldsymbol{\Theta}}_k]_{\pi(i),\pi(j)} \neq 0$ such that

$$-[\boldsymbol{\Theta}^{-1}]_{ij} + S_{ij} - \Lambda_{ij} = -[\widehat{\boldsymbol{R}}_k]_{\pi(i),\pi(j)} + S_{ij} - \Lambda_{ij} = 0. \tag{40}$$

For (37c), the pair $(i,j)$ either satisfies $\psi(i) \neq \psi(j)$ or $\psi(i) = \psi(j)$. For first case, we have $S_{ij} - \Lambda_{ij} \leq 0$ as $(i,j) \notin \text{supp}(\boldsymbol{T})$. Consequently, we obtain

$$-[\boldsymbol{\Theta}^{-1}]_{ij} + S_{ij} - \Lambda_{ij} \leq -R_{ij} \leq 0, \tag{41}$$

as $\boldsymbol{R} \geq \boldsymbol{0}$. For second case where $\psi(i) = \psi(j)$, we have $\Theta_{ij} = [\widehat{\boldsymbol{\Theta}}_k]_{\pi(i),\pi(j)} = 0$. Then, applying (21c) in Lemma A.2, we have

$$-[\boldsymbol{\Theta}^{-1}]_{ij} + S_{ij} - \Lambda_{ij} = -[\widehat{\boldsymbol{R}}_k]_{\pi(i),\pi(j)} + S_{ij} - \Lambda_{ij} \leq 0. \tag{42}$$

Hence, together with the fact that $\boldsymbol{\Theta}$ is primal feasible and positive definite, we conclude that $\boldsymbol{\Theta}$ satisfies the KKT conditions of convex Problem (1). Therefore, $\boldsymbol{\Theta}$ is the optimal solution of Problem (1) and Theorem 3.3 is obtained.

## C    Extensions and Discussions

### C.1    Extensions to traditional graphical lasso

The traditional graphical lasso is formulated as

$$\underset{\boldsymbol{\Theta} \succ \mathbf{0}}{\text{minimize}} \quad -\log \det (\boldsymbol{\Theta}) + \langle \boldsymbol{\Theta}, \boldsymbol{S} \rangle + \sum_{i \neq j} \Lambda_{ij} |\Theta_{ij}| . \tag{43}$$

The KKT conditions of graphical lasso include:

$$-R_{ij} + S_{ij} = 0 \qquad \text{if } i = j, \tag{44a}$$

$$-R_{ij} + S_{ij} - \Lambda_{ij} \cdot \text{sign} (\Theta_{ij}) = 0 \qquad \text{if } i \neq j, \Theta_{ij} \neq 0, \tag{44b}$$

$$-\Lambda_{ij} \leq -R_{ij} + S_{ij} - \leq \Lambda_{ij} \qquad \text{if } i = j, \Theta_{ij} = 0. \tag{44c}$$

In the realm of graphical lasso, the thresholded sample covariance matrix is usually defined as

$$T_{ij} = \begin{cases} 0 & \text{if } |S_{ij}| \leq \Lambda_{ij}, \\ S_{ij} - \Lambda_{ij} \cdot \text{sign} (S_{ij}) & \text{if } |S_{ij}| > \Lambda_{ij}. \end{cases} \tag{45}$$

Our main results Theorem 3.3 can be generalized to graphical lasso with some modifications.

**Theorem C.1.** *Suppose that $\mathcal{P}$ is the bridge-block decomposition of the thresholded graph, $\widehat{\boldsymbol{\Theta}}_k$ is the optimal solution of $k$-th sub-problem, i.e.,*

$$\widehat{\boldsymbol{\Theta}}_k = \arg \min_{\boldsymbol{\Theta}_k \succ \mathbf{0}} -\log \det (\boldsymbol{\Theta}_k) + \langle \boldsymbol{\Theta}_k, \boldsymbol{S}_{\mathcal{V}_k} \rangle + \sum_{i \neq j} \left[ \boldsymbol{\Lambda}_{\mathcal{V}_k} \odot |\boldsymbol{\Theta}_k| \right]_{ij} . \tag{46}$$

*The decomposed form, which is given as*

$$\boldsymbol{\Theta}_{i,j}^{\star} = \begin{cases} [\widehat{\boldsymbol{\Theta}}_k]_{\pi(i), \pi(i)} + \zeta_i & \text{if } i = j \in \mathcal{V}_k, \\ [\widehat{\boldsymbol{\Theta}}_k]_{\pi(i), \pi(j)} & \text{if } i \neq j \text{ and } i, j \in \mathcal{V}_k, \\ -T_{ij} / (S_{ii} S_{jj} - T_{ij}^2) & \text{if } (i, j) \in \mathcal{B}, \\ 0 & \text{otherwise}, \end{cases} \tag{47}$$

*in which $\zeta_i = \frac{1}{S_{ii}} \sum_{(i,m) \in \mathcal{B}} \frac{T_{im}^2}{S_{ii} S_{mm} - T_{im}^2}$ and $\zeta_i = 0$ if $\forall m : (i, m) \notin \mathcal{B}$, achieves optimality if $\boldsymbol{\Theta} \succ \mathbf{0}$ and the following inequality is satisfied:*

$$| - R_{ij} + S_{ij}| \leq \Lambda_{ij}, \quad \forall i \neq j \text{ and } i, j \text{ belong to different clusters.} \tag{48}$$

It is important to note that the decomposed form (47) aligns with the structure of (7), with the key distinction lying in the computation of the thresholded matrix $\boldsymbol{T}$. The proofs of Theorem C.1 can be conducted similar to Section B.

The condition (48) can be guaranteed under some circumstances. For example, we can set large values for $\Lambda_{ij}$ whenever $i \neq j$ and $i, j$ are in different clusters. In this case, the thresholded graph is sparse and the condition (48) is satisfied. Nevertheless, in the majority of practical situations, verifying this condition poses a substantial challenge. Conversely, these conditions are inherently satisfied when MTP$_2$ constraints are applied.

### C.2    Extensions to dense graphs

The proposed method operates under an implicit assumption of sparsity, which is a common characteristic of high-dimensional applications where the focus is on identifying only the most significant relationships among variables. However, it is possible to encounter scenarios where the thresholded graph is dense, making it difficult to identify any bridges for conducting bridge-block decomposition. In such cases, the proposed framework can still be useful, and this section will primarily focus on exploring its potential applications under these circumstances. We will discuss various strategies and techniques for adapting the proposed method to deal with dense graphs, enabling a broader applicability of the proposed bridge-block decomposition approach.

One of typical approaches to make the proposed method applicable is to tune the values of regularization coefficients. As the thresholded matrix is computed as

$$\boldsymbol{T} = \max\left(\boldsymbol{0}, \boldsymbol{S} - \boldsymbol{\Lambda}\right), \tag{49}$$

we can always increase the value of $\Lambda_{ij}$ to make the thresholded graph supp$(\boldsymbol{T})$ sufficiently sparse. When tuning the hyper-parameters is not recommended, a more general strategy is to use the solution of proposed framework as an initial point of numerical algorithms.

**Warm-start algorithm**: One of the main possible strengths of the proposed decomposed-form solution is that it can be treated as an initial point (warm-start) for the numerical algorithms specialized for solving Problem 1. To apply the decomposition approach, we first need to derive a suitable vertex partition of the graph based on the given data matrix $\boldsymbol{S}$ and regularization matrix $\boldsymbol{\Lambda}$. There are several possible vertex partitioning techniques that can be used, including:

- $k$-edge-connected decomposition: This method divides a connected graph into its $k$-edge-connected sub-graphs, which can be useful for identifying bridges or bottlenecks in the graph structure.
- $k$-vertex-connected decomposition: This method decomposes a graph into its $k$-vertex-connected sub-graphs, which can be useful for identifying clusters or communities of vertices that are tightly connected to each other.
- $k$-core decomposition: This method identifies the maximal subgraphs of the graph in which each vertex has a degree of at least $k$, which can be useful for identifying the most densely connected regions of the graph.
- Community detection: This method identifies groups of vertices that are densely connected within the group and sparsely connected to vertices outside the group, which can be useful for identifying clusters or communities of vertices with similar properties or behavior.

Once we obtain a vertex partition $\mathcal{P} = \{\mathcal{V}_1, \ldots, \mathcal{V}_K\}$, where $K$ refers to the number of clusters, we can compute the initial point using the following form:

$$\Theta_{i,j} = \begin{cases} [\widehat{\boldsymbol{\Theta}}_k]_{\pi(i),\pi(i)} + \zeta_i & \text{if } i = j \in \mathcal{V}_k, \\ [\widehat{\boldsymbol{\Theta}}_k]_{\pi(i),\pi(j)} & \text{if } i \neq j \text{ and } i,j \in \mathcal{V}_k, \\ -T_{ij}/(S_{ii}S_{jj} - T_{ij}^2) & \text{if } (i,j) \in \mathcal{E}_e, \\ 0 & \text{otherwise}, \end{cases} \tag{50}$$

in which $\mathcal{E}_e$ refers to a set of external edges among clusters, i.e.,

$$\mathcal{E}_e = \{(i,j) \,|\, (i,j) \in \text{supp}\left(\boldsymbol{T}\right), \psi\left(i\right) \neq \psi\left(j\right)\}, \tag{51}$$

$\zeta_i$ is computed as

$$\zeta_i = \frac{1}{S_{ii}} \sum_{(i,m)\in\mathcal{E}_e} \frac{T_{im}^2}{S_{ii}S_{mm} - T_{im}^2}, \tag{52}$$

and $\widehat{\boldsymbol{\Theta}}_k$ is computed as the optimal solution of $k$-th sub-problem:

$$\widehat{\boldsymbol{\Theta}}_k = \arg\min_{\boldsymbol{\Theta}_k \in \mathcal{M}^{p_k}} -\log\det\left(\boldsymbol{\Theta}_k\right) + \langle\boldsymbol{\Theta}_k, \boldsymbol{S}_{\mathcal{V}_k} - \boldsymbol{\Lambda}_{\mathcal{V}_k}\rangle. \tag{53}$$

*Remark* C.2. The initial point is the optimal solution if the vertex-partition $\mathcal{P}$ is a bridge-block decomposition of the optimal graph.

*Remark* C.3. When the vertex partition $\mathcal{P}$ is given as $\mathcal{P} = \{\{1\}, \{2\}, \ldots, \{p\}\}$, i.e., $\mathcal{V}_k = \{k\}$ for $k = 1, \ldots, p$, the initial point (50) reduces to a closed-form expression:

$$\Theta_{i,j} = \begin{cases} \frac{1}{S_{ii}}\left(1 + \sum_{(i,m)\in\text{supp}(\boldsymbol{T})} \frac{T_{im}^2}{S_{ii}S_{mm} - T_{im}^2}\right) & \text{if } i = j, \\ -T_{ij}/(S_{ii}S_{jj} - T_{ij}^2) & \text{if } (i,j) \in \text{supp}\left(\boldsymbol{T}\right), \\ 0 & \text{otherwise}. \end{cases} \tag{54}$$

In practice, utilizing the warm-start strategy can also enhance the performance of certain numerical algorithms.

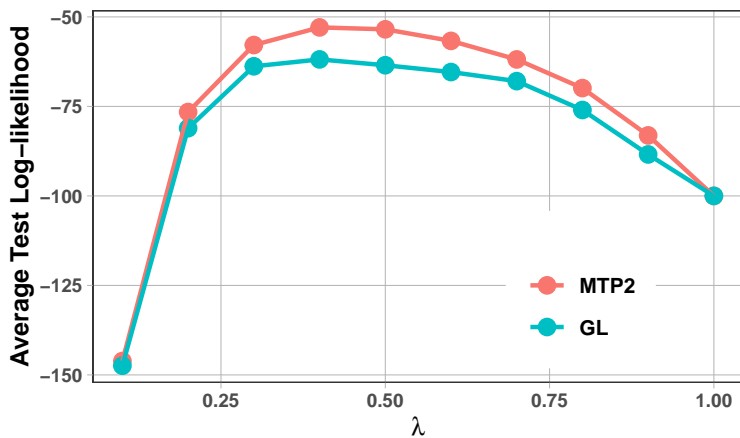

Figure 8: Average log-likelihood in out-of-sample data for different precision matrix estimation models as a function of the sparsity promoting hyperparameter $\lambda$.

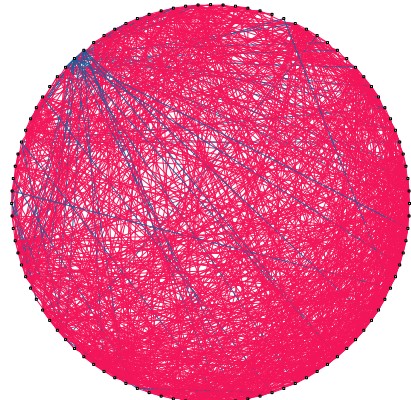

Figure 9: Estimated Graphical Lasso of Crop data set at their highest average likelihood. Red edges represent positive conditional correlations, while blue edges represent negative ones.

## D  Additional Comments on Real-World Experiments

To show that $MTP_2$ assumption approximately holds for Crop dataset, we selected 20 random subsets from the Crop dataset. For each subset, we computed the Graphical Lasso and the $MTP_2$ graphical model for different values of using the first 10 observations. The remaining 36 observations were used to calculate the out-of-sample log-likelihood, which was then averaged across all datasets. This process allows us to evaluate how well these models generalize to unseen data.

As depicted in Figure 8 of the attached PDF, the $MTP_2$ graphical model outperforms the Graphical Lasso, providing a higher test log-likelihood. We present one instance of the estimated graphical lasso model in Figure 9. It reveals that most conditional correlations are positive (red edges, 90%), with a few being negative (blue edges, 10%). This pattern implies strong positive dependence in the Crop data, aligning with the characteristics of $MTP_2$.

These results are not unexpected, given that the Crop dataset comprises multiple clusters. Within the same cluster, we expect data points to exhibit greater similarity compared to those in different clusters. This situation signifies a form of positive dependence, thereby justifying the plausibility of the $MTP_2$ assumption in this problem.

