# OpenReview forum: "Learning Large-Scale MTP$_2$ Gaussian Graphical Models via Bridge-Block Decomposition"
_NeurIPS.cc/2023/Conference — NeurIPS 2023 poster_

### Official Review · Reviewer_pk8s · 2023-07-07

**Soundness:** 3 good
**Presentation:** 2 fair
**Contribution:** 3 good
**Rating:** 6
**Confidence:** 4

**Summary:**

The authors show that, in Gaussian MTP2 distributions, bridges in the graph structure have a closed form solution.  They use this observation to suggest practical solutions that can be applied whenever such models are being fit.

**Strengths:**

A nice, more or less self-contained theoretical work that unifies and extends some existing lines of research into MTP2 Gaussian distributions.

**Weaknesses:**

Some room for improvement in terms of presentation and some minor typos.  It's harder for me to judge practical utility.  While it is true that MTP2 distributions have some practical utility (I've used them myself), it isn't clear that many real datasets meet that requirement (even approximately).

**Questions:**

- "calculating the thresholded graph, bridges, and clusters, is negligible" Sure, but I feel that a precise statement and citation is needed here.  In particular, you still have to compute the sample covariance matrix for this.

- I actually had to read it a couple times to understand the exact procedure that was being proposed.  I suggest adding some clarifying details to help the reader.

-  What is the primary limitation that makes this result only apply to MTP2 distributions?  Is it possible that something similar could be true more generally?  I didn't really get a sense of why MTP2 was required in the main text.

-  Do you need Gaussian distributions here?  Can something be done for more general MTP2 distributions, e.g., those that can still be represented as pairwise graphical models?

Misc. typos (only a few listed):
- "which severs as a common assumption"
- "make possibilities of solving..." (revise)
- "As thresholded graph plays"
- "garnered considerable attentions"
- "Bridge is one of the important concepts"
- "closed-form solutions in literature heavily"
- "seems inapplicable for dense graph"

**Limitations:**

Some discussion of limitations, though it would have been nice to see a more clearly identified set of problems yet to be solved.

---

> ### Author Rebuttal · Authors · 2023-08-09
>
> ## Answer to Questions Part 1
>
> >"calculating the thresholded graph, bridges, and clusters, is negligible" Sure, but I feel that a precise statement and citation is needed here. In particular, you still have to compute the sample covariance matrix for this.
>
> __Reply:__ We are grateful for your valuable suggestion. We would like to elaborate on this matter and provide a more precise statement. For example, given sample covariance matrix, the computational cost associated with calculating the thresholded graph, bridges, and clusters is relatively low [1,2].
>
> For a detailed discussion on why the computational cost associated with determining the thresholded graph, bridges, and clusters is much lower compared to solving sub-problems, please refer to Part 3 of our global response.
>
> __References__:
>
> [1] Jens M Schmidt. A simple test on 2-vertex-and 2-edge-connectivity. Information Processing Letters, 113(7):241–244, 2013.
>
> [2] R Endre Tarjan. A note on finding the bridges of a graph. Information Processing Letters, 2(6):160–161, 1974.
>
> ## Answer to Questions Part 2
>
> > I actually had to read it a couple times to understand the exact procedure that was being proposed. I suggest adding some clarifying details to help the reader.
>
> __Reply:__ We appreciate your suggestion that the procedure could benefit from additional clarification. Please find our detailed answer in the part 3 of the global response.
>
> ## Answer to Questions Part 3
>
> >What is the primary limitation that makes this result only apply to MTP2 distributions? Is it possible that something similar could be true more generally? I didn't really get a sense of why MTP2 was required in the main text.
>
> __Reply:__ We appreciate the reviewer for raising this interesting question. In part 1 of our global response, we highlight the importance of MTP2 in ensuring the validity of our proposed Theorem 3.3. This also explains why our results only apply to MTP2.
>
> In response to whether our methodology can hold more broadly, in part 2 of our global response, we detail the circumstances under which our theoretical framework could be applied to the graphical lasso as well. Please check our detailed answer.
>
> ## Answer to Questions Part 4
> > Do you need Gaussian distributions here? Can something be done for more general MTP2 distributions, e.g., those that can still be represented as pairwise graphical models?
>
> __Reply:__ Thank you for your insightful comment. While our paper primarily focuses on learning $\mathrm{MTP}_2$ Gaussian graphical models, our method is not explicitly tied to Gaussian distributions. This is due to the deterministic nature of our optimization problem, as presented below:
>
> $ \min_{\boldsymbol{\Theta}\succ \mathbf 0} - \log \det ( \boldsymbol{\Theta}) + \langle \boldsymbol{\Theta}, \mathbf{S} \rangle + \sum_{ij} \Lambda_{ij} | \Theta_{ij}|.$
>
> Extending our method to non-Gaussian cases may be straightforward. For instance, in the context of elliptical distributions, we could substitute the sample correlation matrix with the Kendall's tau correlation matrix, leading to positive partial correlation graphs [1]. It's worth highlighting that the MTP2 property is somewhat restrictive, and as demonstrated in [2], an elliptical distribution that is MTP2 across all dimensions is essentially Gaussian. Lastly, we would like to clarify that the sole assumption in our paper is that $S_{ij} < \sqrt{S_{ii} S_{jj}}$ for any $i \neq j$.
>
> We will include a more detailed discussion on this aspect in our revised manuscript.
>
> __References__:
>
> [1] R. Agrawal, U. Roy, and C. Uhler. "Covariance matrix estimation under total positivity for portfolio selection." Journal of Financial Econometrics, 20(2):367-389, 2022.
>
> [2] D. Rossell, and P. Zwiernik. "Dependence in elliptical partial correlation graphs." Electronic Journal of Statistics, 15(2):4236-4263, 2021.
>
> ## Reply to Comments
>
> >  While it is true that MTP2 distributions have some practical utility (I've used them myself), it isn't clear that many real datasets meet that requirement (even approximately)
>
> __Reply:__ Thank you for your thoughtful questions. From a practical perspective, MTP2 becomes a natural choice when the __variables are anticipated to display positive dependence__. Numerous real-world scenarios, particularly in sectors such as finance and social sciences, manifest this positive dependence.
>
> To illuminate how data expressing positive correlation may approximate MTP2 properties, we have conducted supplementary experiments with the CROP Image Data set, which is used in our real-world experiment. The details are described in Part 4 of our global response. These experiments reveal that the CROP data set shows a form of positive correlation, thereby making it suitable for modeling with MTP2 graphical models. Please refer to our global response for our detailed replies.
>
> ## Additional Comments
> __Reply__: Thank you for pointing out the typos. We will correct them in the revised version.

---

### Official Review · Reviewer_ghhX · 2023-07-17

**Soundness:** 2 fair
**Presentation:** 2 fair
**Contribution:** 2 fair
**Rating:** 6
**Confidence:** 4

**Summary:**

This paper studies a graphical lasso problem where the precision matrix is restricted to be symmetric M-matrix and the associated GMRF graph has a special structure. Specifically, the authors consider the situation when the graph allows bridge-block decomposition so that vertices can be partitioned into k parts by cutting k-1 "bridges".  Under this situation, the authors show that the original problem can be decomposed into k subproblems, illustrated in theorems in section 3.2. Since the existing algorithm's complexity quickly increases as the dimensionality increases, decomposition into subproblems offers large computational benefits. Authors compare the performance of four different algorithms, using synthetic and real data, with and without leveraging the bridge-block decomposition and show that bridge-block decomposition provides huge computational benefits if the given graph has many edges that are “bridges”.

**Strengths:**

Authors provide a closed form of the elements of precision matrix corresponding to bridges, and make a connection with existing literature when underlying GMRF graph is an acyclic graph. This method is useful when one wants to solve MTP2 constrained graphical lasso problem with a penalty parameter that allows block-bridge decomposition. This work is resembles the existing literature on graphical lasso (without MTP2 constraint), such as Witten et al (2011); Mazumder and Hastie (2012), where graphical lasso problem can be decomposed to the smaller subproblems when graph has many connected components, and adds a contribution in the context of MTP2 constrained graphical lasso when graph has many bridges.


Witten, D. M., Friedman, J. H., & Simon, N. (2011). New insights and faster computations for the graphical lasso. Journal of Computational and Graphical Statistics, 20(4), 892-900.

Mazumder, R., & Hastie, T. (2012). Exact covariance thresholding into connected components for large-scale graphical lasso. The Journal of Machine Learning Research, 13(1), 781-794.

**Weaknesses:**

Simulation settings and real data examples have some discrepancy from usual (graphical) lasso settings and its motivation. The main goal of graphical lasso is to discover the underlying conditional independency structure from the multivariate Gaussian data with various choice of penalty parameter (aka lasso path diagram). However authors first fix the graph in their settings such as preferential attachment graph and stochastic block model and choose penalty parameter according to the graph. It is understandable in the simulation study setting to show the computational benefits with block-bridge decomposition, but fixing the graph in the real data is not convincing, and also it is not clearly stated why MTP2 graphical lasso is appropriate to this crop image data problem (is it reasonable to assume that crop image data comes from high-dimensional multivariate Gaussian distribution? what is the interpretation of the resulting graph? why MTP2 constraint is necessary / plausible in this problem?)

In practice, graphical lasso is ran under various settings of penalty parameters and chosen appropriately such as cross validation. The existing graphical lasso decomposition (Witten et al (2011); Mazumder and Hastie (2012)) are useful since graphical lasso with high penalty parameter often leads to a graph with many connected components. In similar spirit, I suggest authors to illustrate how often the graph that allows block bridge decomposition appears by the choice of penalty parameter.

**Questions:**

- Does the main result, decomposition into smaller graphical lasso subproblems under block-bridge decomposition, holds without MTP2 constraint?
- If not, please clarify the the role of the MTP2 constraint in the main results; does MTP2 constraint is necessary for the assumption 2.1. to be hold?
- What happens if graph has many connected components (without bridges) in MTP2 constrained graphical lasso problem? Does similar decomposition holds?
- Section 4.1: Sample size $n$ is 10 times the dimension $p$? This setting completely violates the main motivation of graphical lasso where $\ell_1$ penalty has been proposed since MLE does not exist when $n<p$.
- Section 4.3: Shouldn't be data $y_i$ is 24000-dimensional with $i=1,\dots,46$ observations?


**Limitations:**

The main results are meaningful but simulation results and real data settings are less convincing. No potential negative societal impact.

---

> ### Author Rebuttal · Authors · 2023-08-09
>
> ## Answer to Questions Part 1
>
> >Does the main result holds without MTP2 constraint?
>
> __Reply:__ We thank the reviewer for raising this point. The MTP2 constraints are essential prerequisites for our main results.
>
> ## Answer to Questions Part 2
>
> > If not, please clarify the the role of the MTP2 constraint in the main results.
>
> __Reply:__ We appreciate the reviewer for raising this insightful question. Please refer to part 1 of global response for our detailed answer.
>
> > Does MTP2 constraint is necessary for the assumption 2.1. to be hold?
>
> __Reply:__ The MTP2 constraint is not a prerequisite for assumption 2.1. Assumption 2.1 primarily ensures the existence and uniqueness of the optimal solution.
>
> ## Answer to Questions Part 3
>
> > What happens if graph has many connected components (without bridges) in MTP2 constrained graphical lasso problem? Does similar decomposition holds?
>
> __Reply:__ We appreciate your thought-provoking question. Yes, our theorem can also find the exact solution like the existing graphical lasso decomposition. Here's a simple comparison:
>
> |Graph Type|Description|
> |:-|:-|
> |__Single-component__ graph with bridges|Our method works, the __existing method doesn't__.|
> |__Multi-component__ graph with bridges|Our method __works better__ than existing method.|
> |__Multi-component__ graph without bridges|Our method has __equivalent__ effectiveness to the existing method.|
>
> In theory, bridge-block decomposition refers to the components after removing all bridges. So, even in a graph with many components and no bridges, we can still find clusters. This makes bridge-block decomposition versatile, as it can deal with __both connected and disconnected__ sparse thresholded graphs.
>
> ## Answer to Questions Part 4
>
> > Section 4.1: Sample size n is 10 times the dimension p? This setting completely violates the main motivation of graphical lasso.
>
> __Reply:__ Thank you for your thoughtful feedback. We understand the graphical lasso is often used when $p>n$. However, __the aim of our experiments__ is not merely to uphold the statistical rationale of the MTP2 graphical lasso, but to __demonstrate how our proposed theorem can significantly accelerate the learning of high-dimensional sparse MTP2 graphs from an optimization perspective__. Hence, our experimental design is crafted to ensure successful graph structure recovery across all experiments.
>
> In response to your suggestions, we've added new experiments (see Figure 4 in the attached PDF). Here, we set the sample size to $n=0.1p$, where $p$ is the dimension. These experiments show that our method still effectively speeds up convergence due to the sparsity of the underlying structure.
>
> ## Answer to Questions Part 5
>
> > Section 4.3: Shouldn't be data yi is 24000-dimensional with i=1,...,46 observations?
>
> __Reply:__ In the context of graphical models, every node corresponds to a data, while the edges represent the conditional dependencies between the data. As a result, our network comprises 24000 nodes. It is important to clarify that in this scenario, yi does not represent a feature, but rather a signal consisting of 46 observations. This yields a configuration where p=24000 and n=46.
>
> ## Reply to Comments Part 1
>
> > Fixing the graph is not convincing, and also why MTP2 graphical lasso is appropriate to this crop image data problem (what is the interpretation of the resulting graph? why MTP2 constraint is necessary / plausible in this problem?)
>
> __Reply:__ We appreciate your practical question. To alleviate your concerns, we have included more experiments in Part 4 of the global response to show that the MTP2 graphical Lasso is suitable for graph-based clustering with the CROP dataset.
>
> The results show that the CROP data exhibit positive dependency suit for MTP2 assumption. To the estimated graph, edges represent positive conditional dependence between variables. In a clustering context, this often suggests that these interconnected nodes in the estimated graph tend to fall within the same cluster.
>
> Given our observation that the data adheres to MTP2 properties, it would be advantageous to include MTP2 constraints __as prior information to improve the learning of graphical models__. The MTP2 structure also possesses excellent mathematical properties that confer __significant computational advantages__, enabling applying bridge-block decomposition to solve high-dimensional sparse graphs, which can not be solved via existing methods.
>
> ## Reply to Comments Part 2
>
> >  The existing graphical lasso decomposition are useful since graphical lasso with high penalty parameter often leads to a graph with many connected components. I suggest authors to illustrate how often the graph that allows block bridge decomposition appears by the choice of penalty parameter.
>
> __Reply:__ Thank you for your useful comments on practical aspects. To respond, we've done more tests using the same steps as in our synthetic data experiments. We created a random SBM model with 2000 nodes. By adjusting $\lambda$, we checked how the decomposition methods perform under different thresholded graphs. The results are shown in the table below:
>
> |$\lambda$|0|0.01|0.03|0.1|0.15|0.18|0.3|0.5|1|
> |:-|:-:|:-:|:-:|:-:|:-:|:-:|:-:|:-:|:-:|
> |Number of clusters by __existing decomposition__|1|1|1|1|1|__15__|__1681__|__1974__|__2000__|
> |Number of clusters by __bridge-block decomposition__| 1|1|__9__|__402__|__406__|__411__|__1793__|__1978__|__2000__|
>
> We found that:
>
> 1. Our method becomes effective when $\lambda\geq 0.03$, while the existing decomposition method is applicable when $\lambda\geq 0.18$.
> 2. Theoretically, our approach can handle both scenarios: when the graph has bridges or when it has multiple components. Conversely, the existing method can only cope with the latter case.
>
> The results clearly illustrate that __our method exhibits a broader range of applicability__ compared to existing decomposition methods.

---

> > ### Comment · Reviewer_ghhX · 2023-08-13
> >
> > Thanks for the detailed response. I have some additional questions.
> >
> > > Reply: In the context of graphical models, every node corresponds to a data, while the edges represent the conditional dependencies between the data. As a result, our network comprises 24000 nodes. It is important to clarify that in this scenario, yi does not represent a feature, but rather a signal consisting of 46 observations. This yields a configuration where p=24000 and n=46.
> >
> > At the bottom of page 8 (why there are no line numbers?), the current manuscript reads "Our goal is to perform graph-based clustering for the indexed data ${y_1, . . . , y_{24000}}$ using MTP2 GGMs, where $y_i \in \mathbb{R}^{46}$." This sentence implies $p=46$ and $n=24000$, and this is what I asked. Please cross-check with the notations above eq. (1) in page 2.
> >
> > > (Section 2.2) This paper advances prior research in two ways. Firstly, we extend the closed-form solutions beyond
> > the acyclic graph structure to encompass any edge corresponding to a bridge.
> >
> > > (Global response) Our research offers two main theoretical advancements. The first proposes an explicit form for the inverse of $\Theta$.
> >
> > I feel these statements are too strong. As described in Figure 2, when graph allows block-bridge decomposition, what authors show is the decomposition of the big problem into the smaller subproblems, and closed form solutions for off-blockdiagonal entries (orange entries of Fig. 2). Authors claim that this paper "extend the closed-form solutions beyond
> > the acyclic graph structure" of Fattahi and Sojoudi (2019, JMLR, ref. 27). I think this may confuse readers, since what authors show is mainly the decomposition, not the explicit solution that is similar to equation (11) of Fattahi and Sojoudi (2019). I believe authors should use terms like "closed from" or "explicit from" with extra caution in this regard, and I strongly suggest to revise sentence "we extend the closed-form solutions beyond the acyclic graph structure to encompass any edge corresponding to a bridge" and other sentences with similar context.
> >
> > I appreciate authors running additional simulations when $p>>n$. Authors use preferential attachment (Barabasi-albert) graph and stochastic blockmodel in simulation studies to illustrate block-bridge decomposition, which often has many bridges due to its construction. While those graph arises from completely different context (e.g. community detection problems), and I understand this is for the illustrative purposes, but are there any references that using those preferential attachement graph or SBM in the graphical lasso settings like in this paper?
> >
> > > (Section 2.1) This paper considers estimating the precision matrix $\Theta$ given $n$ **independent and identically distributed**
> > observations ${y_1, . . . , y_n}$ that follow an MTP2 Gaussian distribution"
> >
> > > (section 4.3) Though it is not the focus of this paper to discover insights into the estimated MTP2 GGMs for better understanding the inherent nature of the data.
> >
> > I thank authors to run graphical lasso and present additional real data analysis results (fig.5 and fig. 6). However I am still not sure crop image dataset is best suited for the proposed method. Each node corresponds to each pixel of satellite image, and there are $n=46$ images $y_1,\dots,y_{46}$ which are time-varying measurements, illustrating the temporal evolution of the observed area. This implies data are temporally correlated, not iid. How big is the time interval (montly, yearly)? What is being measured? (pixel greyscale value? please clarify and describe at least minimally, not just referring ref.47) Most importantly, how should we interpret the conditional dependency structure of this real data analysis result?
> > If estimated MTP2 GMM does not give any further understanding of the crop image data, I believe this data should not be used in the real data analysis.
> >
> > I have increased my score from 3 to 4 based on the additional results, but still lean to a rejection due to above concerns.

---

> > > ### Author Response · Authors · 2023-08-14
> > >
> > > Thank you for your valuable comments and for raising the score of our paper. We're more than happy to further discuss and address your concerns.
> > >
> > > ## Reply to Post-Rebuttal Questions Part 1
> > >
> > > > The current manuscript reads "Our goal is to perform graph-based clustering for the indexed data $y_1,\dots,y_{24000}$ using MTP2 GGMs, where $ y_i\in\mathbb R^{46}$." This sentence implies $p=46$ and $n=24000$.
> > >
> > > __Reply__: Thank you for pointing out the ambiguity in our manuscript. We apologize for any confusion. To clarify, we propose to revise the sentence as follows:
> > >
> > > "Our aim is to apply graph-based clustering to the time series data that contains 46 observations, denoted as $\mathbf y_1,\dots, \mathbf y_{46}$, where $\mathbf y_i\in \mathbb R^{24000}$."
> > >
> > > This representation aligns with the conventions typically adopted in the realm of graphical models. We hope this modification adequately addresses your concern. We are committed to improving our manuscript based on your feedback.
> > >
> > > ## Reply to Post-Rebuttal Questions Part 2
> > >
> > > > Authors claim that this paper "extend the closed-form solutions beyond the acyclic graph structure" of Fattahi and Sojoudi (2019, JMLR, ref. 27). I think this may confuse readers, since what authors show is mainly the decomposition, not the explicit solution that is similar to equation (11) of Fattahi and Sojoudi (2019).
> > >
> > > __Reply__:  We deeply appreciate your constructive suggestions. We agree with the reviewer's comment that more caution is needed when using the terms "closed-form" and "explicit solution" to avoid mixing up the results from our paper and the paper of Fattahi and Sojoudi.
> > >
> > > To address your concern, we propose to revise the sentence in our manuscript as follows: "While previous studies have offered an explicit solution for $\Theta_{ij}$ in the case of acyclic thresholded graphs, we reveal that the explicit solution for $\Theta_{ij}$ consistently applies to every $(i,j)$ pair acting as a bridge in non-acyclic graphs."
> > >
> > > To provide further clarification, it's noteworthy that the JMLR paper outlines an explicit computation method for deriving each entry of the solution associated with the acyclic graph structure. However, our paper illustrates that the solution can be obtained using a decomposed approach based on block-bridge decomposition. In this approach, the entries corresponding to bridges have an explicit computation method, similar to the results presented in the JMLR paper, and the decomposed subproblems can be solved independently. Importantly, our results are more general: the JMLR paper necessitates an acyclic graph structure, which is a specific case in our study. This follows from the fact that all edges in an acyclic graph are bridges, which have explicit solutions as demonstrated in our theory.
> > >
> > > Your feedback is highly valuable in helping us improve the clarity and precision of our manuscript. We would make revisions accordingly and ensure a detailed comparison of our results with those of Fattahi and Sojoudi's work in our revised manuscript.
> > >
> > > > Are there any references that using those preferential attachement graph or SBM in the graphical lasso settings like in this paper?
> > >
> > > __Reply__: Yes, both the Barabási–Albert (BA) and Stochastic Block Models (SBM) are frequently used in the context of graphical models, including graphical lasso settings. Below, we provide several relevant references:
> > >
> > > __BA graph:__
> > >
> > > 1. Liu, H., & Wang, L. (2017). TIGER: A Tuning-Insensitive Approach for Optimally Estimating Gaussian Graphical Models.
> > >
> > > 2. Ying, J., Cardoso, J. V. de M., & Palomar, D. (2020). Nonconvex Sparse Graph Learning under Laplacian Constrained Graphical Model. Advances in Neural Information Processing Systems, 33, 7101-7113.
> > >
> > > 3. Li, R., et al. (2023). Graph Learning for Latent-Variable Gaussian Graphical Models under Laplacian Constraints. Neurocomputing, 532, 67-76.
> > >
> > >
> > > __SBM graph:__
> > >
> > > 1. Mohan, K., et al. (2014). Node-based learning of multiple Gaussian graphical models. The Journal of Machine Learning Research, 15(1), 445-488.
> > >
> > > 2. Pircalabelu, E., & Claeskens, G. (2020). Community-Based Group Graphical Lasso. The Journal of Machine Learning Research, 21(1), 2406-2437.
> > >
> > > 3. Ying, J., Cardoso, J. V. de M., & Palomar, D. P. (2023). Adaptive Estimation of Graphical Models under Total Positivity. International Conference on Machine Learning.
> > >
> > > BA models play an important role in network science, which generate random scale-free networks using a preferential attachment mechanism such that new nodes tend to link to nodes that have higher degree in the evolution. Scale-free networks are well-suited to model the Internet, the world wide web, protein interaction networks, citation networks, and most social and online networks. Stochastic block models serve as fundamental tools in network science, creating random networks based on community structures, where nodes within the same group are more likely to form connections.

---

> > > ### Author Response · Authors · 2023-08-14
> > >
> > > ## Reply to Post-Rebuttal Questions Part 3
> > >
> > > > Each node corresponds to each pixel of satellite image, and there are $n=46$ images $y_1,\dots,y_{46}$ which are time-varying measurements, illustrating the temporal evolution of the observed area. This implies data are temporally correlated, not iid.
> > >
> > > __Reply__: Thank you for your insightful comment. We agree with your observation that the nodes, representing each pixel of the satellite image, are indeed temporally correlated due to the time-varying nature of the measurements. This introduces a temporal correlation in the data, which inevitably deviates from the assumption of independent and identically distributed (iid) data.
> > >
> > > Despite the inherent temporal correlation in time series data, it's a common practice in the realm of Gaussian graphical models to approximate the data as being independent and identically distributed (i.i.d). This approximation, while not strictly accurate, has been successfully implemented in various studies such as those by Liu et al. (2012), Wang et al. (2020), and Agrawal et al. (2022). These works utilize Gaussian graphical models for financial time series analysis, and despite the fact that stock prices don't perfectly conform to the i.i.d assumption, they are effectively modeled and interpreted within this framework. This demonstrates the practical utility of such models, suggesting that our approach can still provide valuable insights and meaningful results, even when acknowledging the presence of temporal correlation.
> > >
> > > It is worth mentioning that estimating time-varying graphical models is indeed an intriguing research direction in the context of time series. Such an approach becomes particularly relevant when there's a need to understand the evolving interactive relationships among a set of random variables. While this specific aspect is not the primary focus of our current paper, we acknowledge its importance and potential for future research.
> > >
> > > We greatly appreciate your insightful feedback. We plan to incorporate a more detailed discussion on this matter in our revised manuscript. Should you have any further queries or comments, please do not hesitate to reach out to us.
> > >
> > > __Reference__:
> > >
> > > -- Liu, H., Han, F., & Zhang, C. (2012). Transelliptical graphical models. Advances in Neural Information Processing Systems, 25.
> > >
> > > -- Wang, Y., Roy, U., & Uhler, C. (2020). Learning high-dimensional Gaussian graphical models under total positivity without adjustment of tuning parameters. International Conference on Artificial Intelligence and Statistics. PMLR.
> > >
> > > -- Agrawal, R., Roy, U., & Uhler, C. (2022). Covariance matrix estimation under total positivity for portfolio selection. Journal of Financial Econometrics, 20(2), 367-389.
> > >
> > > > How big is the time interval (montly, yearly)? What is being measured? (pixel greyscale value? please clarify and describe at least minimally, not just referring ref.47)
> > >
> > > __Reply__: Images in the dataset are captured at five-day intervals. The recorded spectral information from these images represents variations in "colours" (referring to different spectral bands) for each pixel over the study period. Hence, each data point in this dataset corresponds to a time series of the spectral changes observed at a specific geographical location over time.
> > >
> > >
> > > >  Most importantly, how should we interpret the conditional dependency structure of this real data analysis result? If estimated MTP2 GMM does not give any further understanding of the crop image data, I believe this data should not be used in the real data analysis.
> > >
> > > __Reply__: Thank you for your insightful question.
> > >
> > > The estimated graph is statistically meaningful. We observed that the bulk of edges are located within the same type of crop, while the edges between nodes associated with different crops are relatively sparse. This finding is beneficial for clustering processes and aligns with our anticipations, as a stronger positive dependency is often exhibited within the same class, while the dependency among different classes tends to be significantly weaker.
> > >
> > > In addition to revealing clustering patterns, our graph can reflect more intricate insights. For example, in Figure 6a of our manuscript, we noticed a substantial density of edges between two crop types, 'temporary meadow' and 'pasture' (colored in gold '#B79F00' and cyan '#00BFC4' respectively), indicating a conditional dependency significantly stronger than those found between other categories. This observation aligns with our perception.
> > >
> > > Therefore, the conditional dependency structure that we have inferred possesses the ability to represent the inherent interrelationships among different crops. We believe this provides valuable insight to further understanding of the crop image data. We appreciate your consideration and look forward to any further questions or comments you may have.

---

> > > > ### Comment · Reviewer_ghhX · 2023-08-18
> > > >
> > > > Thanks for the detailed answers. I believe that incorporation of authors comments to the revised manuscript, especially for the real data analysis section, will make a significant improvement of the paper in my view. I increased my score from 4 to 6, provided that detailed description and interpretation of the real data analysis will be added in the revised manuscript. Adding color labels in figure 6(a) would be a plus.

---

> > > > > ### Author Response · Authors · 2023-08-18
> > > > >
> > > > > Dear Reviewer,
> > > > >
> > > > > We greatly appreciate your expert review and the time you've dedicated to improving our manuscript. We are committed to revising our paper in response to your valued feedback.
> > > > >
> > > > > Best regards
> > > > >
> > > > > The Authors

---

### Official Review · Reviewer_bj5B · 2023-07-24

**Soundness:** 3 good
**Presentation:** 2 fair
**Contribution:** 2 fair
**Rating:** 5
**Confidence:** 3

**Summary:**

The paper studies the problem of estimating the precision matrix, which is the inverse of the correlation matrix, of a given Gaussian random vector $y$. The precision matrix $\Theta$ is assumed to satisfy a technical condition called MTP2 which states that $\Theta$ is symmetric and $\Theta_{i,j} \le 0$. This seems to be a well motivated assumption from various applications. The contribution of this paper is a technique for estimating $\Theta$ as follows: given a predicted sparsity pattern on $\Theta$ in the form of a graph $G$ , the natural optimization problem for estimating $\Theta$ can be solved by first solving the optimization problem on smaller 'blocks' and combining them across 'bridges'. They are defined as follows. Bridges are single edge cuts in  $G$ and the resulting connected vertices are called blocks.

**Strengths:**

The paper shows that given a block bridge decomposition, the optimization problem of estimating $\Theta$ can be efficiently solved by first solving the problem on the individual blocks and then combining the solution across bridges. The paper gives an explicit formula for doing so. In the case where $G$ is sparse, this can represent significant computational savings over estimating the entire precision matrix at once. Furthermore, since the work provides a structural theorem, any optimization algorithm can be used in conjunction with their observation.

**Weaknesses:**

I am not familiar with the literature but it seems like a big assumption to know the threshold graph explicitly. What happens if this graph is unknown? It seems to be more natural that the graph is unknown and one must estimate it.

An intermediate setting which also seems interesting is in the case where we know a noisy approximation to the block bridge structure. How do the proposed methods perform under such noisy information? Are the derived formulas robust?

What is the motivation for even assuming the block bridge structure? I can see social networks being one motivation but it would be more convincing if there were experiments on (real) social networks.


**Questions:**

What happens to the SBM experiments if the edge probabilities between the different blocks are increased? The quality of the method should deteriorate as the probabilities increase, since the block structure increasingly deteriorates. It would be interesting to see how much the proposed method can tolerate.

**Limitations:**

No ethical concerns.

---

> ### Author Rebuttal · Authors · 2023-08-09
>
> ## Reply to Comments Part 1
>
> > I am not familiar with the literature but it seems like a big assumption to know the threshold graph explicitly. What happens if this graph is unknown? It seems to be more natural that the graph is unknown and one must estimate it.
>
> __Reply:__ We appreciate your feedback and would like to provide further clarification. Our problem is defined such that we already have access to the sample covariance matrix, denoted by $\mathbf{S}$, and the regularization matrix, represented as $\boldsymbol{\Lambda}$.
>
> __Given these matrices__, we can __precisely__ compute the thresholded matrix, $\mathbf{T}$, using the formula $\mathbf{T}=\max(\mathbf{0},\mathbf{S}-\boldsymbol{\Lambda})$. Hence, the knowledge of the thresholded graph isn't an assumption we're imposing, but rather a natural outcome resulting from the specific optimization problem at hand.
>
> ## Reply to Comments Part 2
>
> > An intermediate setting which also seems interesting is in the case where we know a noisy approximation to the block bridge structure. How do the proposed methods perform under such noisy information? Are the derived formulas robust?
>
> __Reply:__ Thank you for posing this question. In practical applications, samples inevitably carry noise, making it challenging to obtain an accurate estimate of the sample covariance matrix.
>
> However, regardless of the sample size or the degree of sample noise, we can still apply our proposed method to the noisy sample covariance matrix. This is because the step of computing the bridge-block decomposition from the sample covariance matrix is deterministic, and __we can always obtain an optimal and exact solution__ via our proposed framework. As a result, from an optimization perspective, our method is robust.
>
> ## Reply to Comments Part 3
>
> > What is the motivation for even assuming the block bridge structure? I can see social networks being one motivation but it would be more convincing if there were experiments on (real) social networks.
>
> __Reply:__ Thank you for your insightful question. Our motivation in assuming the block bridge structure stems from our focus on sparse graph learning. In sparse graphs, bridge is a commmon feature, and here's why.
>
> One motivating for learning sparse graphs is to enhance interpretability by preserving only the most significant relationships among variables. Each node connects with its most important neighbors, reducing the probabilities for cycle formations. Furthermore, the connectivity in sparse graphs is rather weak due to the limited number of edges, making them easy to separate. As a result, numerous edges in sparse graphs become bridges - these are edges whose removal would create additional components.
>
>
> In the context of social networks, the stochastic block model is a common tool for representing such networks. Notably, social networks often exhibit strong intra-group connections and weaker inter-group ties. This structure may potentially harbor a considerable number of bridges.
>
> While we didn't have the opportunity to apply our methods to social network learning due to time limit, we believe there is potential merit that our methods could contribute to the more efficient learning of these networks.
>
> ## Answer to Questions
>
> > What happens to the SBM experiments if the edge probabilities between the different blocks are increased? The quality of the method should deteriorate as the probabilities increase, since the block structure increasingly deteriorates. It would be interesting to see how much the proposed method can tolerate.
>
> __Reply:__ We appreciate your highlighting of this practical issue. We concur that the effectiveness of the method might decline as the graph grows denser.
>
> In response to your queries, we conducted additional experiments to examine the extent to which we can speed up the convergence of the BCD method for various values of $p'$. Here, $p'$ represents the probability of forming an edge $(i,j)$, where $i$ and $j$ are any two distinct nodes in neighboring communities.
>
> We utilized a SBM graph with $1500$ nodes for this study. The results are illustrated in Figure 2 of the attached PDF and are also encapsulated in the table below:
>
> |Edge formation probability|0.05|0.10|0.15|0.20|0.25|0.30|0.35|0.40|0.45|
> |:-:|:-:|:-:|:-:|:-:|:-:|:-:|:-:|:-:|:-:|
> | Ratio of Improvement |$179.7$|$150.1$|$80.1$|$60.1$|  $31.1$|$16.6$|$3.6$|$3.5$|$3.2$|
>
> Here, Ratio of Improvement refers to how many times we can accelerate the convergence of BCD methods. As expected, a rise in $p'$ increases the chance of multiple edges linking the blocks, which can potentially hamper the efficiency of our approach.
>
> As we address in Section C of our Appendix, while our method is primarily intended for sparsely connected graphs, it retains its usefulness even for dense graphs. Our method can serve as an approximate solution or provide a warm start for other numerical algorithms, thereby boosting their computational efficiency.

---

> > ### Comment · Reviewer_bj5B · 2023-08-17
> > **Response to rebuttal**
> >
> > Thank you for responding to the review. It seems that knowing $\Lambda$ is still an assumption. For now I will maintain my score.

---

> > > ### Author Response · Authors · 2023-08-18
> > >
> > > Dear Reviewer,
> > >
> > > We appreciate your feedback.
> > >
> > > In practice, the regularization matrix $\boldsymbol{\Lambda}$ can be effectively computed based on certain initial estimates. In our paper, we suggest the use of Nie et al.'s (2016) method for efficiently deriving these initial estimates.
> > >
> > > __Reference__:
> > >
> > > — Feiping Nie, Xiaoqian Wang, Michael Jordan, and Heng Huang. The constrained Laplacian rank algorithm for graph-based clustering. In Proceedings of the AAAI Conference on Artificial Intelligence, volume 30, 2016.

---

### Official Review · Reviewer_vWNe · 2023-07-26

**Soundness:** 3 good
**Presentation:** 4 excellent
**Contribution:** 3 good
**Rating:** 7
**Confidence:** 3

**Summary:**

This paper studies the problem of learning Gaussian Graphical Models (GGMs) satisfying a certain positive associativity condition among the variables, namely that the precision matrix has nonnegative off-diagonal elements. This condition is known as being "multivariate totally positive of order two", or MTP$_2$, and has applications in ML (where it corresponds to attractive Markov random fields), finance, and more.

MTP$_2$ GGMs come with the benefit that the traditional optimization procedure used to estimate the precision matrix of a GGM, namely the graphical Lasso, takes on a particularly simple and smooth form. Prior work had shown various polynomial-time convergence guarantees for the graphical Lasso under the MTP$_2$ assumption, but these are still not very suitable for practical applications (scaling with the dimension $p$ as $O(p^3)$ or $O(p^4)$). Other prior work had shown a closed-form solution for the graphical Lasso under the assumption that the "thresholded sample covariance graph" (an object defined in terms of the sample covariance and the regularization parameters used in the graphical Lasso), or thresholded graph for short, is acyclic.

The main contribution of this paper is essentially to generalize the latter closed-form result in terms of the "bridge-block decomposition" of the thresholded graph. The bridge-block decomposition of a graph is essentially a partition of the graph into components connected only by "bridge" edges. Formally, a bridge edge is an edge such that deleting it increases the number of connected components in a graph; it is effectively "the only edge" bridging two different components (see Fig 1). The paper's main theorem (Thm 3.3) essentially says the following: to solve the graphical Lasso for an MTP$_2$ GGM, compute the bridge-block decomposition of the thresholded graph, run the graphical Lasso for each component separately, and stitch them together using a simple closed-form formula. Moreover, this theorem also readily recovers as a special case the prior closed-form result for acyclic thresholded graphs. This is because in an acyclic graph, every edge is a bridge, and the bridge-block decomposition is particularly simple.

Thus the main result amounts to a divide-and-conquer recipe for learning MTP$_2$ GGMs, and the authors show various numerical experiments suggesting the practical superiority of this method over all prior methods (which operate on the entire graph). A key benefit is that the subproblems may be solved using any graphical Lasso implementation whatsoever, and potentially in parallel.





**Strengths:**

Disclaimer: I am not very familiar with the literature in this area, and my review should be taken as that of a relative outsider.

The paper's main result is both an interesting structural result about MTP$_2$ GGMs as well as a genuinely practical algorithmic advance in learning such models. From a conceptual point of view, the idea of leveraging the bridge-block decomposition seems novel. The overall result seems like a useful and nice contribution to the literature on this problem, and to the extent that one considers MTP$_2$ GGMs significant, one should consider this result significant as well.

The paper is largely clear and easy to follow (modulo some occasionally confusing bits; see the Questions section). It does a good job of setting up the main problem as well as the necessary context. I did not manage to verify the proofs in detail, but they seemed fairly clean, relying on an analysis of the KKT conditions of the graphical Lasso as well as some clever algebraic manipulation.

**Weaknesses:**

I think the main things to really evaluate about this paper are its novelty and significance. As an outsider to this area, I find this hard to accurately gauge, but I think the paper scores well on these fronts.

I do think the paper could benefit from a better conceptual overview of the main proof and the role of the bridge-block decomposition. The context and benefits are discussed adequately, but the key ideas in the proof do not come through very well, and the main proof seemed slightly magical to me. Why would one have expected the bridge-block decomposition to help? Was its role surprising?


**Questions:**

I think the biggest question I have is about the conceptual role of the bridge-block decomposition, as explained above. A few other questions:
- One of the key parts of the eventual proof of Thm 3.3 is Lemma A.2, which is simply described as following from the KKT conditions. Presumably the authors mean the KKT conditions associated with Problem (5)? Since it plays a relatively important conceptual role, I feel this could definitely use more elaboration.
- What exactly is the time required to compute the bridge-block decomposition? This is described as negligible in Sec 3.1, but is it e.g. $O(p^2)$? Similarly, what about the quantities used in the closed-form formula? Currently the only discussion of the overall asymptotic running time appears in the first bullet of the "Proposed Framework" list on page 5.
- In the experimental section, it would be helpful to say more about it is natural to synthesize/define $\Theta$ and $\Lambda$ in the specific way described in the first two paragraphs of Section 4.1, especially for readers who are unfamiliar with the prior work mentioned in those paragraphs.

A couple other nits regarding the presentation:
- It is not immediately obvious how Corollary 3.5 follows from Thm 3.3, and this could use a line of explanation.
- The notation $\mathbb{S}^p$ is not formally defined in the paper, and MTP$_2$'s expansion is only mentioned in the abstract.

**Limitations:**

The paper could use a couple additional lines in the final section about the technical limitations of this work and what the major next steps could be. I am not aware of any significant potential negative societal impact of this theoretical work.

---

> ### Author Rebuttal · Authors · 2023-08-09
>
> ## Answer to Questions Part 1
> > One of the key parts of the eventual proof of Thm 3.3 is Lemma A.2, which is simply described as following from the KKT conditions. Presumably the authors mean the KKT conditions associated with Problem (5)?
>
> __Reply:__ Thank you for your insightful comment. Here, we elaborate on their relationship.
>
> Let's denote $\Gamma_{ij}$ as the dual variables associated with the constraints $\Theta_{ij}\leq 0$.  With $\mathbf{R}=\boldsymbol{\Theta}^{-1}$, the KKT conditions include (1) $-\mathbf{R}+\mathbf{S}-\boldsymbol{\Lambda}+\boldsymbol{\Gamma}=\mathbf{0}$, (2) $\Theta_{ij}\leq0,\forall i\neq j,$ (3) $\Gamma_{ij}\geq0,\forall i\neq j,$ (4) $\Theta_{ij}\cdot\Gamma_{ij}=0,\forall i\neq j.$
>
> We can eliminate the dual variables as follows:
>
> 1. For $\Theta_{ij}<0$, the complementary slackness leads to $\Gamma_{ij}=0$, which implies $-R_{ij}+S_{ij}-\Lambda_{ij}=0$;
>
> 2. When $\Theta_{ij}=0$, we have $\Gamma_{ij}=R_{ij}-S_{ij}+\Lambda_{ij}\geq 0$, which indicates that $-R_{ij}+S_{ij}-\Lambda_{ij}\leq 0$.
>
> Following these steps, we arrive at the optimal conditions for original problem. As the sub-problems have the same form with the original problem, Lemma A.2 holds accordingly. We will incorporate these details in the revised version.
>
> ## Answer to Questions Part 2
>
> > What exactly is the time required to compute the bridge-block decomposition? Is it e.g. O(p^2)? What about the quantities used in the closed-form formula?
>
> __Reply:__ We sincerely appreciate the reviewer's question. Please find our detailed answer in the Part 3 of our global response.
>
> ## Answer to Questions Part 3
>
> > In the experimental section, it would be helpful to say more about it is natural to synthesize/define $\boldsymbol{\Theta}$ and $\boldsymbol{\Lambda}$.
>
> __Reply:__ We appreciate your valuable recommendation and will revise our paper accordingly. The methods we use to synthesize $\boldsymbol{\Theta}$ and $\boldsymbol{\Lambda}$ are guided by [1] and can be explained as follows.
>
> We synthesize $\boldsymbol{\Theta}$ as $\boldsymbol{\Theta}=1.05\cdot\lambda_{\max}(\mathbf{A})\cdot\mathbf{I}-\mathbf{A}$, where $\mathbf{A}$ is the adjacency matrix of the underlying graph. This ensures that $\boldsymbol{\Theta}$ is a positive definite matrix with off-diagonal elements being negative, making $\boldsymbol{\Theta}$ a randomly generated M-matrix. Then, we normalize $\boldsymbol{\Theta}^{-1}$, thereby deriving a randomly generated correlation matrix.
>
> Following this, we set $\Lambda_{ij}=\chi\big/(\epsilon+\Theta_{ij}^{(0)})$. Here, a high penalty is expected on $\Theta_{ij}$ if its initial estimate $\Theta_{ij}^{(0)}$ is small. By applying this way, we can recover the underlying structure by selecting an appropriate value of $\chi$.
>
> [1] Martin Slawski and Matthias Hein. Estimation of positive definite M-matrices and structure learning for attractive Gaussian markov random fields. Linear Algebra and its Applications, 473:145–179, 2015.
>
> ## Answer to Questions Part 4
> > It is not immediately obvious how Corollary 3.5 follows from Thm 3.3.
>
> __Reply:__ We appreciate your valuable suggestion. Here's a clarification on why a bridge $(i,j)$ in the thresholded graph will persist as a bridge in the optimal graph.
>
> An edge $(i,j)$ is a bridge if and only if there exists exactly one unique path connecting nodes $i$ to $j$, denoted as $d_{ij}=\{(i,j)\}$. According to the definition of a bridge, the removal of it would lead to an increment of graph's components. The presence of any additional paths would contradict this definition.
>
> As the optimal graph is a subset of the thresholded graph and $\Theta_{ij}\neq 0$, we find that $d_{ij}=\{(i,j)\}$ continues to be the unique path from nodes $i$ to $j$, which indicates that $(i,j)$ remains a bridge.
>
> ## Answer to Questions Part 5
>
> > The notation $\mathbb{S}^p$ is not formally defined in the paper, and MTP$_2$'s expansion is only mentioned in the abstract.
>
> __Reply:__ We appreciate your keen observation that calls for clarity. The symbol $\mathbb{S}^p$ represents the set of symmetric matrices with a dimension of $p\times p$. We will incorporate the full form of MTP2 into the body of the paper.
>
> ## Reply to Comments
>
> > I do think the paper could benefit from a better conceptual overview of the main proof and the role of the bridge-block decomposition. Why would one have expected the bridge-block decomposition to help? Was its role surprising?
>
> __Reply:__ We sincerely value your comprehensive review of our paper and your engagement with the core concepts underlying our proof.
>
> Our research is influenced by the study of (Fattahi et al., 2019), which indicates that a closed-form solution exists for large-scale graphical lasso problems. Their intriguing findings hold potential for large-scale data sets, yet their theorem imposes some hard-to-validate conditions and requires the thresholded graph to be acyclic.
>
> In the realm of the MTP2 graph learning problem that interests us, we aimed to conduct related research and unearth similar findings. Our investigations led us to discover that (1) the existence of a closed-form solution hinges not on the acyclicity of the thresholded graph, but on whether an edge is a bridge; (2) Given that bridges provide closed-form solutions, if our aim is to decompose the problem, we should form the sub-problem excluding the bridges, giving birth to our bridge-block decomposition strategy; (3) By referencing the proofs in (Fattahi et al., 2019), we eventually prove that all conditions that are typically difficult to verify are naturally satisfied in MTP2 graphical models.
>
> As our approach does not impose additional conditions, and it does not require the thresholded graph to be acyclic, our method has a wider range of applicability.
>
> -- Salar Fattahi and Somayeh Sojoudi. Graphical lasso and thresholding: Equivalence and closed-form solutions. Journal of Machine Learning Research, 2019.

---

> > ### Comment · Reviewer_vWNe · 2023-08-13
> >
> > I am satisfied with the responses, thank you. It would be great to incorporate some of this into the final revision.

---

> > > ### Author Response · Authors · 2023-08-14
> > >
> > > Dear Reviewer,
> > >
> > > We sincerely appreciate the time and effort you've dedicated to reviewing our paper. Your constructive comments have been instrumental in refining our paper.
> > >
> > > Thank you once again for your invaluable input and support.
> > >
> > > Best regards,
> > >
> > > The Authors

---

### Official Review · Reviewer_tnT4 · 2023-07-27

**Soundness:** 3 good
**Presentation:** 3 good
**Contribution:** 3 good
**Rating:** 6
**Confidence:** 3

**Summary:**

The paper focuses on the problem of learning large-scale Gaussian graphical models (GGMs) that are multivariate totally positive of order two (MTP2). The high-dimensional, sparse MTP2 GGMs are not easily manageable due to their size and complexity. The authors propose a novel approach, introducing the concept of a "bridge", to optimize the entire problem into several smaller, more manageable sub-problems and a set of closed-form solutions. The approach is based on the bridge-block decomposition of the thresholded sample covariance graph, which leads to reductions in computational complexity and improvements in existing algorithms.

**Strengths:**

The proposed bridge-block decomposition framework on Gaussian graphical models seems novel. The problem is motivated nicely, and according to the authors, the framework could significantly reduce computational and memory cost.

The proposed method seems to subsume various network structures, including the BA graph and the SBM, which are common models used in network analysis.

Experimental results are provided, and the computational results look promising.

**Weaknesses:**

As the authors mentioned in the paper, the proposed method might not generalize to dense cases. Still I feel in many settings like BA and SBM, sparsity is a reasonable assumption.


**Questions:**

Is there any way to characterize the effect of the proposed approach in terms of some global graphical properties, for example, the edge expansion? Asking this because bridge is related.

**Limitations:**

See Above.

---

> ### Author Rebuttal · Authors · 2023-08-09
>
> ## Answer to Questions
>
> > Is there any way to characterize the effect of the proposed approach in terms of some global graphical properties, for example, the edge expansion? Asking this because bridge is related.
>
> __Reply:__ We appreciate the reviewer's interesting and insightful question. Indeed, certain global graphical properties could potentially serve as indicators of a graph's connectivity strength, thereby providing a measure of our method's effectiveness. However, the computation of edge expansion presents a significant challenge, especially for high-dimensional graphs.
>
> As an alternative, we opted to utilize another global graphical property known as __algebraic connectivity__. This metric, the second smallest eigenvalue of a graph's Laplacian matrix, offers a depiction of the graph's overall connectivity. It can be employed to characterize the impact of our proposed solution.
>
> To demonstrate this, we executed tests on two random SBM graphs with distinct algebraic connectivities. We evaluate the how many times our proposed method can accelerate the convergence of existing algorithms. The corresponding results are displayed in the table below:
>
> |Algebraic connectivity |BCD|PGD|FPN|PQN-LBFGS|
> |:-:|:-:|:-:|:-:|:-:|
> | 4e-5 |$2064$ | $314$  |$213$  |$57$   |
> | 4e-4 |$26$  | $9$  |$13$  |    $5$  |
>
> It is obvious that our proposed method is much more effective on graph with low algebraic connectivity.
>
> Details regarding convergence (refer to Figure 3) and more in-depth analyses (refer to Figure 1) are available in the attached PDF. In detail, we generate numerous random graphs and for each, we examine the relationship between its algebraic connectivity and a theoretical number of how many time we can accelerate a BCD method. In Figure 1, each point plotted represents a specific graph, with the x-axis indicating the algebraic connectivity and the y-axis representing the acceleration factor.
>
> __Conclusion__: The results indicate a general trend: __as the algebraic connectivity decreases, the effectiveness of our proposed method enhances.__
>
> Remarks: The theoretical number of acceleration is computed as follows. Let's assume that a problem with dimension $p$ can be addressed by BCD algorithm in $c\cdot p^4$ seconds, where $c$ is a constant. By implementing the bridge-block decomposition, this cost is reduced to $\sum_kc\cdot p_k^4$, with $p_k$ denoting the size of the $k$-th cluster. Therefore, we define the ratio $p^4\big /\sum_k p_k^4$ as the theoretical number by which our method can speed up the BCD algorithm.
>
> ## Reply to Comments
>
> > The proposed method seems to subsume various network structures, including the BA graph and the SBM, which are common models used in network analysis. As the authors mentioned in the paper, the proposed method might not generalize to dense cases. Still I feel in many settings like BA and SBM, sparsity is a reasonable assumption.
>
> __Reply:__  We appreciate the reviewer's comment and acknowledge that there are numerous applications beyond the scope of this paper where the graph is dense.
>
> In response to this, we have added Section C in our Appendix, which discusses various alternative strategies for integrating our proposed method into the learning of dense MTP2 graphical models. For instance, though our decomposition formula only yields exact solutions for sparse graphs via bridge-block decomposition, it could be regarded as __an approximate solution__ for dense graphs. Additionally, the explicitly decomposed form could serve as an __effective warm start__ for other numerical algorithms.

---

### Author Rebuttal · Authors · 2023-08-09

## Part 1: Answers to Questions Regarding Roles of MTP2

> We gather this question from Reviewers ghhX and pk8s, who sought the role of the MTP2 constraint in our main results and why these only apply to MTP2 distributions.

__Reply__: We thank the reviewers for bringing up this interesting point. Our research offers two main theoretical advancements. The first proposes an explicit form for the inverse of $\boldsymbol{\Theta}$, while the second confirms that $\boldsymbol{\Theta}=\mathbf{R}^{-1}$ meets optimality without extra conditions.

__The MTP2 properties are sufficient conditions for the second contribution to hold__. Hence, while the bridge-block decomposed form of $\mathbf{R}$ could be broadly applied to other graphical models, our main findings are only applicable to MTP2 graphical models.

Technically speaking, the MTP2 constraints eliminate the non-smoothness, thus __simplifying the KKT conditions__ as follows:

||Graphical Lasso|MTP2|
|:-|:-|:-|
|$\forall i$| $-R_{ii}+S_{ii}=0,$|$-R_{ii}+S_{ii}=0,$|
|$\forall\Theta_{ij}\neq0$|$-R_{ij}+S_{ij}+\lambda_{ij}\text{sign}\left(\Theta_{ij}\right)=0,$| $-R_{ij}+S_{ij}-\lambda_{ij}=0$|
|$\forall\Theta_{ij}=0$|$\vert -R_{ij}+S_{ij}\vert\leq \lambda_{ij}$|$-R_{ij}+S_{ij}-\lambda_{ij}\leq0$|

Simultaneously, $\mathbf{R}$ becomes non-negative ($\geq\boldsymbol{0}$). As a result, The most challenging part $-R_{ij}+S_{ij}-\lambda_{ij}\leq 0$ of the KKT conditions holds under MTP2 properties. (The details refers to Section B of our Appendix.)

## Part 2: Answer to Questions Regarding Generalizing Our Results to Graphical Lasso

> Reviewers ghhX and pk8s provided feedback regarding the applicability of our findings to the graphical lasso.

__Reply__: We thank reviewers posing this thought-provoking question. In the context of the graphical lasso, our current, yet-to-be-published research indicates that our decomposed form achieves optimality if the following inequality is satisfied:

$| -R_{ij} +S_{ij}|\leq \lambda_{ij},\quad\forall (i,j)\notin \mathcal B\text{ and }i,j \text{ belong to different clusters,}$

where $\mathbf{R}$ is computed by Theorem 3.4. This condition could be easily established in certain cases, such as when $\lambda_{ij}\gg 1$. Nevertheless, in the majority of practical situations, verifying this condition poses a substantial challenge.

Conversely, these conditions are inherently satisfied when MTP2 constraints are applied. We hope that our theorem will instigate future studies to simplify these conditions for the graphical lasso.

## Part 3: Answers to Questions Regarding the Exact Procedures of Proposed Framework

> We receive this question from Reviewers vWNe and pk8s, who seek to comprehend the exact procedures and processing time of the proposed method.

__Reply__: We'll use an SBM graph with p=5000 as a reference. The specifics are outlined in the following:

__Preprocessing__: In this phase, we have three steps:

|Step|Elements to Compute|Time (s)|
|:-:|:-|:-:|
|1| Derive the thresholded matrix $\mathbf{T}$ from $\mathbf{S}$ and $\boldsymbol{\Lambda}$ |0.01|
|2|Extract the set of all bridges $\mathcal B$ from the thresholded graph |0.21|
|3|Determine the bridge-block decomposition|0.12|

__Solving Subproblems__: For all clusters, we solve the associated sub-problems.

__Computing Optimal Solution via Theorem 3.3__: The optimal $\boldsymbol{\Theta}$ is then derived as follows:

|Conditions|Formulas|Cost|
|:-|:-|:-|
|$i,j\in \mathcal V_k$| Derive $\Theta_{ij}$ from $\widehat{\boldsymbol{\Theta}}_k$|Depends on specific algorithms (BCD: 967s; FPN: 101s).|
|$(i,j)\in\mathcal B$|$\Theta_{ij}= -T_{ij}/(S_{ii}S_{jj}-T_{ij}^2)$|0.12s|
|otherwise|$\Theta_{ij}=0$|0s|

The findings reveal that the additional costs, such as the time for pre-processing and the employment of Theorem 3.3, are considerably less compared to the time invested in solving sub-problems.

We find bridges using a bridge-finding algorithm from [1]. This method uses a depth-first search, which means the complexity is $\mathcal{O}(|\mathcal V| +|\mathcal E|)$. In the sparse graphs we're interested in, the number of edges $|\mathcal E|$ usually scales similarly to the number of nodes $|\mathcal V|$. Hence, the computational cost of bridges for high-dimensional sparse graphs is low.

[1] R Endre Tarjan. A note on finding the bridges of a graph. Information Processing Letters, 2(6):160–161, 1974.

## Part 4: Experiments on Testing MTP2 in CROP Dataset

> In our paper, we mainly focus on how proposed method acclerate the learning of MTP2 graphical models on CROP Data set. It is suggested by the reviewer ghhX that we should also justify why MTP2 constraint is plausible in this problem. To address this, we present additional experiments to tackle this concern.

__Reply__: We selected 20 random subsets from the CROP dataset. For each subset, we computed the Graphical Lasso and the MTP2 graphical model for different values of $\lambda$ using the first 10 observations. The remaining 36 observations were used to calculate the out-of-sample log-likelihood, which was then averaged across all datasets. This process allows us to evaluate how well these models generalize to unseen data.

As depicted in Figure 5 of the attached PDF, the MTP2 graphical model outperforms the Graphical Lasso, providing a higher test log-likelihood. We present one instance of the estimated graphical lasso model in Figure 6. It reveals that __most conditional correlations are positive__ (red edges, 90%), with a few being negative (blue edges, 10%). This pattern implies strong positive dependence in the CROP data, aligning with the characteristics of MTP2.

These results are not unexpected, given that the CROP dataset comprises multiple clusters. Within the same cluster, we expect data points to exhibit greater similarity compared to those in different clusters. This situation signifies a form of positive dependence, thereby justifying the plausibility of the MTP2 assumption in this problem.

---

### Decision · Program_Chairs · 2023-09-21

**Decision:**

Accept (poster)

**Comment:**

This paper studies methods to more efficiently learn large Gaussian graphical models from data, specifically in the context of the MTP2 (attractiveness/positive correlation) assumption and by attempting to take advantage of inherent 'bridge' structure in the data. This is a fairly specific setting (for example, the MTP2 assumption is reasonably well-motivated but probably not exactly satisfied in most real life datasets), but after the discussion overall the reviewers appreciated the contribution including the experimental validation. Based on this, I recommend acceptance.